# Commercially Available Cell-Free Permeability Tests for Industrial Drug Development: Increased Sustainability through Reduction of In Vivo Studies

**DOI:** 10.3390/pharmaceutics15020592

**Published:** 2023-02-09

**Authors:** Ann-Christin Jacobsen, Sonja Visentin, Cosmin Butnarasu, Paul C. Stein, Massimiliano Pio di Cagno

**Affiliations:** 1Department of Physics, Chemistry and Pharmacy, University of Southern Denmark, 5230 Odense, Denmark; 2Department of Molecular Biotechnology and Health Sciences, University of Turin, 10124 Turin, Italy; 3Department of Pharmacy, Faculty of Mathematics and Natural Sciences, University of Oslo, Sem Sælands Vei 3, 0371 Oslo, Norway

**Keywords:** PAMPA, PermeaPad^®^, permeability, unstirred water layer, dissolution-permeation, in vitro AUC, biowaiver

## Abstract

Replacing in vivo with in vitro studies can increase sustainability in the development of medicines. This principle has already been applied in the biowaiver approach based on the biopharmaceutical classification system, BCS. A biowaiver is a regulatory process in which a drug is approved based on evidence of in vitro equivalence, i.e., a dissolution test, rather than on in vivo bioequivalence. Currently biowaivers can only be granted for highly water-soluble drugs, i.e., BCS class I/III drugs. When evaluating poorly soluble drugs, i.e., BCS class II/IV drugs, in vitro dissolution testing has proved to be inadequate for predicting in vivo drug performance due to the lack of permeability interpretation. The aim of this review was to provide solid proofs that at least two commercially available cell-free in vitro assays, namely, the parallel artificial membrane permeability assay, PAMPA, and the PermeaPad^®^ assay, PermeaPad, in different formats and set-ups, have the potential to reduce and replace in vivo testing to some extent, thus increasing sustainability in drug development. Based on the literature review presented here, we suggest that these assays should be implemented as alternatives to (1) more energy-intense in vitro methods, e.g., refining/replacing cell-based permeability assays, and (2) in vivo studies, e.g., reducing the number of pharmacokinetic studies conducted on animals and humans. For this to happen, a new and modern legislative framework for drug approval is required.

## 1. Introduction

The development and production of new medicines requires the use of cutting-edge technology. Not surprisingly, the pharmaceutical industry is a high-intensity CO_2_ emitter. Recent analyses have shown that the CO_2_ emissions of the pharmaceutical industry even exceed those of the automotive industry [1] or are at least on the same level [2]. Research aiming to increase sustainability in the pharmaceutical industry, e.g., through the use of cleaner production methods and green materials, is a step in the right direction [3]. Nevertheless, some areas of sustainability research remain to be expanded, e.g., studies on waste management and on the impact of new, expensive treatments such as gene therapy [3].

### 1.1. In Vivo Studies in Humans and Animals—A Challenge for Sustainability

An overlooked activity in the pharmaceutical industry which involves relatively high CO_2_ emissions is the performance of in vivo animal and human studies [4]. The carbon footprint of in vivo clinical trials can be reduced through the use of more efficient study designs to limit trial-related travel [4]. Taking this a step further, another obvious way to reduce the CO_2_ emissions associated with clinical trials is to replace them with less CO_2_-intense methodologies, i.e., in vitro studies. Clearly, clinical trials cannot be replaced in their entirety as they are the only way to determine the efficacy of a treatment in patients.

The introduction of drug regulatory approval, based on in vitro evidence, i.e., biowaivers, can be considered as a first significant step towards the reduction of the number of in vivo studies [5]. The BCS classifies drugs according to their aqueous solubility and permeability into four different classes, i.e., classes I–IV, Figure 1. According to the International Conference on Harmonization (ICH) M9 guideline on BCS-based biowaivers [6], a drug is considered highly soluble if its highest dose can be dissolved in 250 mL aqueous media over the pH range of 1.2–6.8 at 37 °C ± 1 °C. A drug is considered highly permeable if its absolute bioavailability is >85%. Alternatively, permeability can be assessed using cell-based in vitro permeability experiments [6].

It is important to underline that cell-based permeability experiments suffer from practical limitations, such as limited standardization and reproducibility, e.g., it is impossible to control the absolute gene expression of the cells [7]. Furthermore, cell studies suffer from poor time- and cost-effectiveness, which is related to the preparation procedure. Cell lines need to be cultured in dedicated sterile cell-lab facilities for 4 to 21 days, depending on the cell line and its application, before use [8]. These limitations might have indirectly promoted the routine implementation of in vivo bioavailability studies in recent years.

The BCS-based biowaiver approach was specifically designed to reduce the need for in vivo bioequivalence studies, i.e., it can provide a surrogate to assume in vivo bioequivalence. A waiver for in vivo bioequivalence studies can be granted for orally administered immediate-release, solid dosage forms and suspensions containing highly soluble drugs, i.e., drugs from BCS classes I and III, with systemic action [6]. This means that with the current framework, biowaivers cannot be granted for drug products containing poorly soluble drugs, i.e., BCS class II and IV drugs, and thus in vivo studies cannot be circumvented in these cases. BCS class II and IV drugs make up a large proportion of approved drugs, which has grown steadily over time (see later Figure 3 in Section 2). Furthermore, biowaivers cannot be granted if the drug product contains excipients that may affect drug absorption [6]. Expanding the BCS-based biowaiver approach to those drugs holds great potential in regard to reducing the number of in vivo studies.

In terms of sustainability, another overlooked activity in the pharmaceutical industry is the performance of in vivo animal studies. It is commonly acknowledged that animal breeding (for food) involves high CO_2_ emissions [9]. Even though less information is available on the carbon footprint of animal breeding and in vivo animal studies in the pharmaceutical industry, the environmental impact of animal testing can still be considered substantial [10].

Animal testing is the practice of using animals to test the safety and efficacy of drugs intended for human use. Animal testing remains the gold standard in pharmaceutical research and development (R&D), from early drug discovery to the formulation testing phase.

Rodents, i.e., rats and mice, are the most frequently implemented animals, especially in early phases. Due to the higher costs and more complex ethical considerations involved, larger animals such as dogs, (mini-)pigs, and non-human primates are more often used in the later stages. The crucial parameters that are determined by animal testing are the drug’s toxicological profile; its pharmacodynamics, e.g., the minimal effective concentration; and its pharmacokinetics, e.g., the fraction absorbed (%A), maximum blood concentration (C_MAX_), and time of peak plasma concentration (t_MAX_).

It is very difficult to obtain the absolute numbers reflecting the real magnitude of animal testing worldwide. Looking only at the European Union (EU), the latest report published by the Commission (Figure 2) showed that in 2015 and 2016, almost 10 million animals were sacrificed as part of in vivo testing. From 2016 to 2017, this number decreased slightly [11]. This slight decrease was followed by a significant increase in 2018 [12].

In contrast to the EU, there is no reliable way to determine exactly how many animals are used in research in the United States. The U.S. Department of Agriculture only maintains a count of research animals protected by the Animal Welfare Act. Therefore, rats, mice, and birds are not accounted for, despite being the species that are most commonly used for testing. Various associations for the protection of animal rights estimate that 100 million animals are sacrificed each year in the U.S. One of the latest estimations was made by Taylor and Alvarez [13], who gave a comprehensive global figure of 200 million animals sacrificed each year for R&D purposes.

Animal testing raises sustainability concerns as well as ethical and scientific concerns. The 3Rs principle, i.e., Replacement, Reduction, and Refinement, was developed to provide a framework for more ethical animal research. On paper, the protection of animals, as guided by the 3Rs principle, is crucial for both the U.S. Food and Drug Administration (FDA) and the European Union and is recorded in law, i.e., Directive 2010/63/EU.

From the perspective of scientific outcomes, is animal testing in pharmaceutical research useful for understanding complex biopharmaceutical problems such as human oral drug absorption? A growing body of scientific literature critically assessing the validity of animal experimentation has raised major concerns about the reliability and predictive value of animal testing for human outcomes and for understanding human physiology [14].

### 1.2. Replacing In Vivo Studies with In Vitro Studies

In vitro biopharmaceutical experiments hold great promise in regard to replacing, or at least reducing, in vivo studies. In vitro assays have seen a steady increase in both their quality and quantity in the last two to three decades. According to Goh et al. [15], from 1980 to 2013 there was a large and steady increase in the use of in vitro tests by both pharmaceutical companies and contract research organizations in all fields, including ADME, i.e., absorption, distribution, metabolism, and excretion; genotoxicity; and safety and pharmacology.

In the BCS-based biowaiver approach, in vivo bioequivalence studies are replaced with in vitro dissolution studies, a procedure that is accepted both by the FDA and the European Medicines Agency (EMA), and which is described in detail in the ICH M9 guideline on BCS-based biowaivers. In vitro dissolution tests are straightforward and easy to standardize in comparison to in vitro permeation studies, which has aided their widespread industrial implementation. However, drug dissolution is only one in a series of steps required for oral drug absorption. Another crucial step is drug permeation. As pointed out in a comprehensive review conducted by the UNGAP network, i.e., COST action 16,205, dissolution studies are often not suited to describe the fate of advanced formulations in the gastrointestinal (GI) tract, which has contributed to the insufficient implementation of in vitro studies in drug development [16], as dissolution alone cannot account for drug permeability.

Cell-free permeation assays (see Berben et al. (2018) for a comprehensive review [17]) have gained increasing interest in pharmaceutics. For industrial implementation, commercially available cell-free permeation assays are especially suitable since their production is standardized and this contributes to the reproducibility of results. Cell-free permeation assays are most often employed to determine the permeability of active pharmaceutical ingredients (APIs) and new chemical entities (NCEs) in early-stage drug development. They could, in principle, be used for a time-/cost-effective BCS classifications of drugs and new chemical entities. Artificial biomimetic barriers have also been used in dissolution/permeation assays [16]. This application of cell-free permeability barriers could potentially expand the BCS-based biowaiver approach towards BCS class II and IV drugs.

Over the years, some theoretical misconceptions linked to these assays have somehow taken root among a part of the scientific community (see Section 3.1.1 and Section 3.1.2 below). This unfortunate fact has somehow put the full-scale deployment of such powerful and promising tools “on hold”.

With this review, we intend to provide an honest picture of the state of the art regarding the cell-free in vitro permeability assays that are commercially available. To provide a theoretical foundation, detailed information on the basics of permeability testing is provided, before the commercially available cell-free permeability assays are reviewed. Finally, combined dissolution/permeation models applying commercially available cell-free permeation assays are presented.

## 2. Industrial Implementation of In Vitro Permeability Testing

### 2.1. Methodology of the Search

To obtain an overview of the current implementation of in vitro permeability testing in the pharmaceutical industry (Table 1), we conducted a literature search using the Scopus search engine. In our search methodology, we identified the 11 largest pharmaceutical companies worldwide according to their annual revenue in 2021 (see Table 1). Next, we used Scopus to identify the total number of publications from those companies through Scopus’ “Affiliation” search function, i.e., searching for the company name. As exclusion criteria, we refined the results by searching for the topic of “in vitro permeability testing”. For the analysis of the commercial products, product overviews (see Appendix A) were obtained from the companies’ official homepages. Worldwide product lists were used when available. Otherwise, overviews of products on the U.S. market were used. Sinopharm was not included in the analysis since a product overview was not available online. The products were divided into non-oral and oral products. Furthermore, the oral products were divided into BCS class I/III products, which were eligible for biowaivers, and BCS class II/IV products, which were not eligible for biowaivers. The year of approval and each drug’s BCS class were determined using official documents from the FDA, EMA and the World Health Organization (WHO), which were available online. The outcomes of the analysis of commercial products are presented in Figure 3.

### 2.2. Results of the Search

Although not absolute, it is reasonable to assume that the number of publications related to a specific topic should be, to some extent, proportional to the economic resources invested in the topic and the trust given to the topic. Clearly, the scientific activities of large pharmaceutical companies are very broad. Still, the current picture drawn on the basis of Table 1 seems discouraging, as only one company, i.e., AbbVie, scored above 1% in terms of its results/total number of publications listed on the topic of ”in vitro permeability testing”. Referring to Figure 3, AbbVie was the company that had the largest share of oral products within their portfolio, i.e., 65% of their products were oral products. This observation may be related to AbbVie’s apparent interest in permeability testing. Johnson & Johnson was the company that scored second in terms of its total number of publications on the topic of ”in vitro permeability testing”, with 0.9%. Similarly, Johnson & Johnson had the second largest share of oral products within their portfolio, i.e., 63% of their products were oral products. In contrast, companies with a small share of oral products within their portfolio, e.g., Sanofi and Pfizer, scored low in terms of the number of publications. Analyzing the BCS classes of the oral products showed that, on average, 50% of oral products were BCS class II/IV drugs (indicated in red in the pie charts in Figure 3). Especially, BCS class II/IV dominated the portfolios of Janssen and Merck (~65%). When sorting by the year of approval, Figure 3b shows that the number of BCS class II/IV drugs increased over time. Indeed, the results presented in this section are prone to uncertainty as the search was based only on published data and marketed drug products. Thus, the analysis cannot provide insights into the development procedures of drugs that are current in the pipeline or into their properties, e.g., their BCS class or drug type. Nevertheless, the analysis provided a substantial picture of the challenge that the pharmaceutical industry is facing, e.g., an increasing number of poorly soluble drugs, which are more challenging to characterize in vitro. The lack of a modern legislative framework has contributed to these challenges.

## 3. In Vitro Flux Studies and Permeability Quantification

Biopharmaceutics is routinely used in pharmaceutical R&D to generate data that can be used to predict in vivo performance. When it comes to oral drug delivery, an efficient in vitro biopharmaceutical assessment should be able to link an empirical parameter to in vivo pharmacokinetic parameters measured in animals or in humans, i.e., providing in vivo–in vitro correlation, IVIVC.

In the past, it was believed that the partition/distribution coefficient could serve the purpose of estimating oral bioavailability. However, it is now evident that partition/distribution coefficients are not suited for this purpose as they do not sufficiently correlate with the oral fractions absorbed by humans/animals in a consistent manner [19]. Approximately 30 years ago, the concept of *apparent permeability,* P_app_, was introduced in pharmaceutics [5]. Within the pharmaceutical context, permeability can be defined as the ability of a chemical entity, e.g., API and NCE, to pass through a biological or biomimetic barrier [20].

As depicted in Figure 4, drug molecules can cross the intestinal epithelium by means of different mechanisms, which can be either passive or active. Transcellular and paracellular diffusion (see Figure 4a(A,B)) are driven by passive diffusion, i.e., the drug is transported along the concentration gradient. In passive transport, the drug is transported along the concentration gradient, whereas in active carrier-mediated transport, the drug transport rate is independent of its concentration and it requires the consumption of energy by cells. In the context of carrier-mediated transport (Figure 4a(C–E)), it is also important to underline the role of efflux transporters that decrease drug absorption by transporting the drug from the cell back into the lumen against the concentration gradient (Figure 4a(E)) [14]. Transcytosis is another transport mechanism that is mediated by vesicles (Figure 4a(D)).

Originally, in the BCS system, Amidon et al. correlated the rate of drug absorption across the intestinal epithelium with the human permeability, P_h_. P_h_ was introduced as a derived coefficient, calculated on the basis of the human fraction absorbed, A%, of a drug compound [5]:(1)$$P_{h}=\frac{j_{h}}{c_{h}}$$
where j_h_ is the human intestinal flux of the drug, i.e., A% normalized by the estimated GI surface area, and c_h_ is the drug concentration at the intestinal wall interface [5]. Permeability was therefore introduced not as an empirically measured in vitro parameter, but its assessment was based on the extent of absorption derived from human pharmacokinetic studies.

Today, permeability is commonly determined by means of in vitro flux studies. Permeability assays are conducted in diffusion cell systems consisting of two water-tight compartments, generally called a donor and an acceptor, which are made from glass or plastic, and which are separated by a barrier that is designed to be biomimetic. A schematic representation of diffusion cell systems is presented in Figure 5, where the barrier is represented by the yellow segment. The ideal in vitro permeability assay for a freely dissolved compound should give P_app_ values that correlate linearly with in vivo human permeability, P_h_ (Figure 4b), and exponentially with the human fraction absorbed, A%, as shown in Figure 4c [21].

Diffusion cell systems can be either vertical or horizontal in terms of their set-up (Figure 5). One of the most frequently employed vertical diffusion cell systems (Figure 5a) is known as Franz cells, e.g., those commercially available from PermeGear Inc. and Logan instruments Corp. A common type of horizontal diffusion cell is side-by-side cells (Figure 5c). Even though both vertical and horizontal set-ups are routinely employed, the possibility of stirring in both the donor and acceptor compartment represents an advantage of the horizontal set-up over the vertical set-up. Multiwell-plate systems (Figure 5b) have been developed, which have a vertical orientation, similarly to the Franz cell set-up. Multi-well plate systems are especially suitable for high-throughput screening (HTS) applications. Additionally, in vitro permeability assays can be classified into two sub-categories according to the type of barrier used, i.e., cell-based and cell-free barriers.

In terms of their data output, the main difference between cell-based (the first assay ever introduced was the immortalized cell line of human colorectal adenocarcinoma cells permeation test, Caco-2 [22,23]) and cell-free approaches is that cell-free assays account for and measure only the passive diffusion of the drug molecule (Figure 4a(A,B)), whereas cell-based assays, depending on cell type, also account for carrier-mediated transport and transcytosis (Figure 4a(C–E)). Passive diffusion occurs for all drugs and in all circumstances, with different magnitudes [21], whereas carrier-mediated transport and transcytosis takes place only in specific cases, which means that not all drugs are subject to carrier-mediated transport and transcytosis [24].

In the scientific community, there is an ongoing debate regarding which transport mechanism contributes the most to drug absorption and distribution in vivo [25,26]. As an example, in 2011 Kell et al. presented a paradigm-breaking review paper, in which they claimed that drug absorption was promoted only by carrier-mediated transport mechanisms [27]. One of the strongest arguments, on which Kell and co-workers based their assumptions, was the “discrepancies in magnitudes of 100-fold or more” between the permeability determined in different in vitro permeability assays and in vivo permeability.

In 2012, a review paper written by the most prominent experts worldwide in the field of oral drug absorption indicated that passive diffusion was the major absorption pathway [25]. Since this is a thermodynamic process, it is quite safe to assume that passive diffusion is the major, and in some cases the only [26], contributor to drug absorption and distribution in the human body. However, one of the points raised by Kell et al. is yet to be addressed: classical indicators of passive transport, i.e., partition coefficients and P_app_, are not always aligned with in vivo data. It is reasonable to believe that these discrepancies are linked to the non-standardized nature of the interpretation of permeability data, as well as misunderstandings of passive diffusion mechanisms, rather than the unaccountability of carrier-mediated transport. In the first part of this review, we present the limitations and common misinterpretations involved in in vitro permeability models.

### 3.1. Conventionally Accepted Physical Models of Passive Permeability

#### 3.1.1. Simplified Homogeneous One-Phase Model

In biopharmaceutical research, it is commonly accepted that Fick’s first law of diffusion (Equation (2)) is a sufficiently good approximation to describe passive permeation, i.e., the flux (j) through a barrier is expressed as:(2)$$j=-D\times \frac{\mathrm{dc}}{\mathrm{dx}}$$
where D is the diffusion coefficient of the drug through all the space (X) and dc/dx represents the concentration gradient within the donor and acceptor compartments (Figure 6).

This model assumes that the diffusion of drug molecules through a heterogeneous system, i.e., one in which different physical phases and interfaces need to be crossed, can be well approximated by means of a homogeneous diffusion model (Figure 6). In the case in which the activity coefficient depends weakly on the concentration, the drug spontaneously diffuses through the barrier following a constant gradient of concentration according to Equation (3):(3)$$j=-D\times \frac{c_{d}-c_{a}}{h_{b}}$$
where j is the flux, c_d_ and c_a_ are the drug concentration in the donor and the acceptor, h_b_ is the thickness of the barrier, and D is the diffusion coefficient of the drug through the space, which is assumed to be constant over X.

Furthermore, assuming that the concentration of the donor is significantly higher than that of the acceptor (c_d_ >> c_a_), Equation (3) can be further simplified as:(4)$$j=D\times \frac{c_{d}}{h_{b}}$$

For this model to fit, however, three assumptions must be made:The drug must have homogeneous and constant c_d_ and c_a_ values, i.e., a constant concentration gradient;The interface transition kinetic within the water phase and the barrier should be irrelevant; andD must be a constant all over space x.

The standard way to empirically assess permeability is to quantify the amount of the drug accumulated over time in the acceptor compartment (also called the receiver) and to calculate the flux, j, as represented in Figure 7 [20].

As a matter of convenience, the permeated mass, Q, e.g., the weight or number of moles, is often normalized over the surface area available for permeation, A, and plotted over time, t. The mass flux, j, of the compound, i.e., the slope of the linear regression in Figure 7a, through the biomimetic barrier is than calculated with Equation (5) as follows:(5)$$j=\frac{Q}{A\times t}$$

P_app_ is calculated by normalizing j over c_d_ as follows:(6)$$P_{\mathrm{app}}=\frac{j}{c_{d}}$$

According to the condition chosen, e.g., if sink conditions are assured throughout the whole experiment, it is also commonly accepted that a measurement at the end-point (generally at 6 h) can be sufficient for the determination of the P_app_ (Figure 7a—end-point slope). In this case, an alternative equation (Equation (7)) proposed by Sawada et al. [28] is commonly accepted:(7)$$P_{\mathrm{app}}=\frac{V_{d}}{{A*Q}_{d}\left(0\right)}\times \left(\frac{\Delta Q_{a}\left(t\right)}{\Delta t}\right)$$
where A is the area available for permeation; V_d_ is the volume of the donor solution; ΔQ_a_(t) is the change in mass of the compound that accumulated in the receiver solution over a time interval, Δt, when the flux was linear and sink conditions were met; and Q_d_(0) is the initial amount of the compound.

It is worth mentioning that Sawada and co-workers called the permeability “effective” (P_eff_) but for consistency in this article, we converted the name into the one that is currently most frequently employed, P_app_. Using one single endpoint for the calculation of P_app_ is very convenient from a practical point of view, but it might lead to the misinterpretation of data in cases where (i) sink conditions and a steady state are not assured; (ii) the lag time is significant; or (iii) non-linear flux is observed, e.g., in the case of enabling formulation testing.

#### 3.1.2. Advanced Multiphasic Model with Interfaces

A more precise and complete physical model to interpret in vitro permeation accounts for the presence of unstirred water layers in both the donor, UWL_d_, and the acceptor, UWL_a_, close to the barrier (Figure 8), as well as physical interfaces, i.e., phase boundaries, between the aqueous and lipid barrier phases [29]. This model, also known as the inhomogeneous solubility-diffusion model, was successfully applied by Diamond et al. in 1974 to study the diffusion of nonelectrolytes through water–lecithin interfaces [30]. According to this model, assuming that that diffusion through the barrier can be described by the Nernst−Planck equation, in the absence of activated processes causing abrupt changes in the solute’s chemical potential or mobility at the water/membrane interfaces, the following expression for the steady-state flux of solute molecules through the barrier applies:(8)$$j=\frac{c_{L}-c_{-L}}{\int_{-L}^{L}\frac{1}{K\left(X\right)\times D_{B}\left(X\right)}}\mathrm{dx}$$
where K and D_B_ are the partitioning and the diffusivity in the barrier of the migrating compound as a function of the position, X. The integration limits represent the boundaries of the barrier, i.e., ±L (Figure 8). It is therefore possible to define the barrier resistivity, R_B_, i.e., the resistance opposed by the barrier to solute permeation, and its reciprocal function expresses the barrier permeability, P_B_, i.e., the permeability through the barrier, as in Equation (9) [31]:(9)$$R_{B}=\frac{1}{P_{B}}=\int_{-L}^{L}\frac{1}{K\left(X\right)\times D\left(X\right)}\mathrm{dX}$$

If we consider the layers involved in the permeation process, i.e., UWL_d_, the barrier and UWL_a_, as flow units in series, we can treat them as resistors in series; therefore, the total resistivity, i.e., R_TOT_, which is the reciprocal function of apparent permeability, P_app_, can be defined by adapting the equation from Savada et al. [28] as follows:(10)$$R_{\mathrm{TOT}}=R_{\mathrm{Ud}}+R_{B}+R_{\mathrm{Ua}}=\frac{1}{P_{\mathrm{Ud}}}+\frac{1}{P_{B}}+\frac{1}{P_{\mathrm{Ua}}}$$
where R_Ud_, R_B_, and R_Ua_ are the resistances of the UWL_d_, of the barrier, and of the UWL_a_, with P_Ud_ and P_Ua_ indicating the permeability though the UWL_d_ and UWL_a_, respectively. Equation (9) can be combined with Equation (10) giving Equation (11):(11)$$R_{\mathrm{TOT}}=\frac{h_{\mathrm{Ud}}}{D_{w}}+\int_{-L}^{L}\frac{1}{K\left(X\right)\times D\left(X\right)}\mathrm{dX}+\frac{h_{\mathrm{Ua}}}{D_{w}}$$

In this case, the model describing permeability is much more comprehensive as it takes into consideration the presence of stagnant water layers on both sides of the barrier, i.e., UWL_d_ and UWL_a_, which is assumed to maintain a constant thickness throughout the whole experiment, i.e., h_Ud_ and h_ua_, respectively [32].

In both UWLs, the concentration of the compound is not constant, but it decreases with space, i.e., X. In these layers the resistance to permeation is solely defined by the diffusivity of the drug in water, D_w_, and the layer thicknesses, i.e., h_Ud_ and h_Ua_. From Equation (11) it is clear that the permeation process depends on the partitioning properties of the drug, i.e., K, but it can also be heavily influenced by the drug’s diffusivity in water. Moreover, in this model, the real concentration gradient that should be taken into account is the concentration difference between the solute in each of the UWLs next to the barrier, i.e., at the interfaces [33].

Although the influence of the stagnant liquid layers on permeability is often overlooked, it is very important and worthy of discussion. For example, Avdeef and co-workers investigated the role of the stagnant liquid layers in the parallel artificial membrane permeation assay (PAMPA) model (discussed below) [34]. They found a clear correlation within the efficacy of the stirring conditions and the unstirred water layer permeability, P_u_. Specifically, they verified empirically that efficient stirring in the acceptor reduces the thickness of the UWL_a_, increasing P_u_ for desipramine, imipramine, propranolol, and verapamil. Brewster and co-workers demonstrated the same effect for griseofluvin, carbamazepine, and hydrocortisone, with the same cell-free permeability set-up [35]. Korjamo et al. showed, using a cell-based permeability model (i.e., the Caco-2 model), that reducing the thickness of the UWL_a_ from 830 µm down to 300 µm by applying different stirring rates ranging from 250 rpm up to 430 rpm enhanced the P_app_ of propranolol by a factor of 2.5 [36].

Eriksen and co-workers compared orbital shaking and magnetic stirring as agitation methods for permeability studies using PermeaPad multiwell-plates (see Section 4.2). They showed that the permeability of albendazole, a BCS class II drug, increased with increased stirring rates but not with increasing shaking rates. Using a dye, they also visually demonstrated that better mixing, which reduced the thickness of the UWL, was achieved with magnetic stirring as compared to orbital shaking [37]. Also using PermeaPad multiwell-plates, Tzanova et al. measured the P_app_ of hydrocortisone and caffeine in the presence of different polyethylene glycols, i.e., PEG400 up to PEG3000, and different PEG concentrations [38]. They clearly demonstrated that the viscosity of the donor solution played a crucial role in P_app_ measurements and that for caffeine, the UWL_d_ represented the major obstacle to permeation, rather than the barrier itself.

Considering the observations presented in this section, it becomes clear that the success of in vitro permeability testing in drug discovery and development depends on the careful interpretation of data, the use of optimal assay conditions and implementation and integration strategies, and the education of users. Moreover, a successful cell-free permeation assay should be highly biomimetic, i.e., it should be able to account for both partitioning kinetics and diffusivity phenomena as biological barriers in vivo.

## 4. Commercially Available Biomimetic Cell-Free In Vitro Permeability Assays

### 4.1. The Parallel Artificial Membrane Permeation Assay (PAMPA)

PAMPA is the most commonly implemented cell-free permeability assay. It was introduced in 1998 in a four-page communication letter that briefly summarized the promising results obtained by Kansy et al. at F. Hoffmann-La Roche laboratories in Basel (CH) [39]. It can be safely stated that PAMPA is, de facto, an industrially developed technology which has found implementation in drug development and academia.

The original PAMPA was designed in a 96-well plate format (schematized in Figure 5b), in which the hydrophobic filter material, i.e., Durapore/Millipore with a pore size of 0.22–0.45 µm was soaked with a 1–20% solution of lecithin in an organic solvent (Figure 9, dodecane, hexadecane, 1,9-decadiene). Kansy and co-workers compared the measured flux (not the P_app_) of 25 drugs in the PAMPA at two different pH values (6.5 and 7.5) with human absorption values (A%). The correlation of the in vitro and in vivo parameters was promising, showing a similar trend to that of partitioning in vivo. A very good correlation of PAMPA permeability and permeability measured in biological cell models, i.e., Caco-2 and in situ gut rat models, was also proven [40].

The original PAMPA model, as well as the associated data interpretation approaches, were subsequently optimized to increase the in vitro–in vivo correlation. For example, Sugano et al. used a mixture of phosphatidylcholine (0.8%), L-α-phosphatidylethanolamine (0.8%), L-α-phosphatidylserine (0.2%), L-α-phosphatidylinositol (0.2%), cholesterol (1.0%), and 1,7-octadiene (97.0%) in the barrier [41]. They tested 33 structurally diverse compounds at different pH values, i.e., from 5.5. to 7.4, and with different donor media. The major results of this study can be summarized as follows. (i) This composition gave the barrier a better in vitro–in vivo correlation (IVIVC), in comparison to the original barrier. (ii) The barrier was proven to sustain dimethyl sulfoxide (DMSO) and PEG400 up to 30%. (iii) Organic solvents such as ethanol could lead to barrier degradation and the overestimation of permeability. Notably, in this work, Sugano at al. used the apparent permeability, P_app_, which they called “the permeability coefficient through the artificial membrane—P_am_”, as an empirical parameter, which was defined as:(12)$$P_{\mathrm{app}}=-2.303\frac{V_{d}\times V_{a}}{V_{d}+V_{a}}\times \frac{1}{A\times t}\log\left(1-\frac{j\%}{100}\right)$$
where V_d_ and V_a_ are the donor and acceptor volumes, respectively; A is the surface area available for permeation; t is the time of incubation; and j% is the flux in percentage resulting from the ratio of the optical densities, i.e., the molar attenuation coefficients, of the acceptor compartment and the reference compartment, i.e., plane buffer, at the end of the experiment.

In the same year, Avdeef et al. tested a simpler lipid mixture of a 2% 2-Dioleoyl-sn-glycero-3-phosaphocholine dodecane solution, which was deposited over a porous microfilter disc [42]. They measured the permeability of a series of lactones and verified the barrier stability over time by testing different stirring rates. Their most important finding was that the experimental time could be reduced from 15 h, as originally defined by Kansy et al. [39], to approximately 6 h for moderately/highly permeable compounds. Moreover, they showed that stirring at more than 500 rpm could hamper the barrier stability over time.

Later studies showed that varying the type of lipids in the PAMPA barrier could greatly affect the permeability outcome. For example, the team of Professor Polli tested a series of different phospholipids, including two phosphatidylcholines, two phosphatidylethanolamines, and two phosphatidylserines. They found that the permeability of some compounds, e.g., metoprolol, was greatly affected by the lipid composition of the PAMPA barrier [43]. These findings opened up avenues of research that focused on developing tissue-specific versions of the PAMPA barrier. Ottaviani et al. introduced a PAMPA barrier mimicking the skin, in which porous polyvinylidene fluoride (PVDF) filters were impregnated with a silicon oil/isopropyl myristate of a different grade [44]. They found that when using the two components in a ratio of 70:30, it was possible to achieve an acceptable correlation within human skin permeability and apparent permeability measured through the PAMPA-skin model (R^2^ = 0.81). The correlation was even better if only the permeability of the compound through the stratum corneum was taken into account (R^2^ = 0.93). To mimic the blood–brain barrier (BBB), Di et al. impregnated the porous filter support with a polar brain lipid mixture with phosphatidylethanolamine (33%) and cerebrosides and sulfatides (30%) as its main constituents (from Avanti Polar Lipids, Inc., Birmingham, AL, USA) and compared the results with a PAMPA barrier consisting of the commercially available lipid mixture [45]. The PAMPA-BBB results correlated well with the standard PAMPA, especially for predicting compounds with lower BBB absorption. For compounds with higher BBB absorption, PAMPA-BBB proved to be a slightly better predictor, whereby false negatives could be avoided.

In another application, PAMPA has been successfully used to predict the in vivo performance of drugs in the presence of solubilizing excipients. Dahan et al. [46] and Brewster et al. [35] successfully employed PAMPA for predicting the in vivo reduction in absorption of lipophilic drugs, i.e., progesterone and hydrocortisone, in the presence of cyclodextrins. Dahan et al. later demonstrated that PAMPA could improve the understanding of the role of solubilizers in enhancing/reducing in drug absorption [46]. Beig et al. showed that PAMPA successfully predicted the increased in vivo absorption of etoposide as amorphous solid dispersion formulations containing D-α-tocopheryl polyethylene glycol succinate (vitamin E TPGS) [47].

PAMPA has proven to be an efficient tool to estimate the permeability of drugs and, to some extent, to enable the use of formulations. From an academic perspective, PAMPA’s compositional flexibility, i.e., the fact that filter material and lipid mixture can be easily altered, and the simplicity of its preparation process, i.e., the fact that the filter is impregnated with the lipid mixture, are advantages. From an industrial perspective, however, those advantages may be limitations. For applications requiring a high degree of standardization, the need for manual impregnation of the filter with the lipid mixture before the experiment is a drawback as it takes time and may introduce some variability to the experiment, e.g., human handling, as well as lab-to-lab and site-to-site variations. Nevertheless, it should be mentioned that standardized lipid mixtures and 96-well filter plates are commercially available, e.g., via Sigma-Aldrich. Furthermore, a ready-to-use precoated version of PAMPA has been developed [48] and is also commercially available (Corning BioCoat precoated PAMPA plate system).

The structure of the commercially available precoated PAMPA differs from that of the original PAMPA as it is composed of three layers. However, its ready-to-use format makes it better suited for industrial applications. The precoated PAMPA has especially found applications in medicinal chemistry projects [49,50,51,52], even including protein testing [53]. Furthermore, the precoated PAMPA has been used to determine the pH-dependent solubility and permeability profiles of >20 compounds [54] and it was used to calibrate an in silico molecular dynamics simulation method to study permeability. Recently, the precoated PAMPA was used to construct a highly biomimetic mucus-PAMPA model, which was used to study pulmonary drug absorption with a focus on cystic fibrosis [55]. In this study, the P_app_ of 45 structurally diverse compounds were measured through the precoated PAMPA barrier, which was coated with a special form of mucus resembling that of cystic fibrosis patients. As expected, for most of the compounds, higher viscosity of the mucus in the donor compartment reduced the permeability of the compounds. However, for eight compounds, i.e., mostly acidic drugs, including ketoprofen, indomethacin, piroxicam and diclofenac, the permeability increased in the presence of the mucus when calcium ions also were present. These results were in accordance with tests performed with human sputum samples demonstrating the impressive biomimetic features of this model.

The PAMPA barrier shows incompatibility with some excipients, especially above certain thresholds. In a comprehensive review, Berben et al. [17] summarized incompatibilities of different PAMPA types with solubilizers such as Cremophor EL (0.5%), Tween 80 (0.5–1 mg/mL), narrow-range ethoxylate (Brij) 35 (>1 mg/mL), Solutol HS 15 (>1 mg/mL), and Triton-X. The reason for this instability seems to reside in the barrier structure (schematic representation depicted in Figure 9). In PAMPA, the lipid mixture that fills the filter pores is directly exposed to the donor environment, allowing for solubilization of the (phospho)lipids by aggressive surfactants that are present in the media. This phenomenon can be worsened by stirring and with time. On the other hand, PAMPA has proven to be stable at any pH, making this assay the ideal tool for pH gradient experiments and electrolyte-drug absorption studies. As proof of this, Yamauchi and Sugano [56,57] demonstrated that PAMPA could mimic and predict the in vivo incompatibility effects between tetracycline and divalent ions in the gastrointestinal tract. The PAMPA barrier seems unable, due to its structural design, to account for paracellular transport. PAMPA has shown a satisfactory correlation with partition coefficients, but it has somehow failed to correctly interpret the permeability of small hydrophilic compounds. For instance, Wexler et al. measured very low permeabilities for caffeine [58], which is known to have a very rapid and complete oral absorption [59]. Recently, de Souza Teixeira et al. showed that according to PAMPA permeabilities determined using a Franz cell set-up, caffeine and theophylline should be classified as medium-permeable compounds [60]. These reports contradict in vivo human data showing that both xanthines are completely and rapidly absorbed, i.e., 100% bioavailable [59]. The reason for this incongruence probably lies in the kinetics that control the permeation through the PAMPA. As depicted in Figure 9, the rate-limiting step in the permeation through the PAMPA barrier is the partition/distribution of the drug molecules from the water into the lipoidal environment that fills the filter pores. This supports a reasonable correlation between PAMPA permeabilities and logP [61]. However, this barrier structure does not allow for small hydrophilic compounds to take advantage of their chemical spaces, e.g., their small size and hydrophilic nature, as would occur in cell lines or in vivo, where small hydrophilic compounds can cross the biological barriers through the paracellular pathway [62]. However, this issue is relevant only for a limited number of specific APIs and NCEs. There are several molecular descriptors that have been found to be predictive of permeability on PAMPA. The most relevant of these are the lipophilicity, i.e., partition/distribution coefficients; logP/logD; the molecular weight; and the polar surface area (PSA) [63]. It is important to note, however, that the best descriptor for predicting permeability on PAMPA may vary depending on the specific dataset, the dataset bias, and the model used for analysis [64].

### 4.2. PermeaPad

The PermeaPad technology was developed at the University of Southern Denmark (Odense, Denmark) in 2013 by Di Cagno and Bauer-Brandl [65]. In contrast to PAMPA, which was introduced on a 96-multiwell plate format, PermeaPad was first developed for classical diffusion cell systems (Figure 5a,c). Originally, the PermeaPad was hand-crafted by depositing phospholipids, e.g., phosphatidylcholine, between two support sheets, e.g., cellulose hydrate, giving the PermeaPad a “sandwich” structure. The manufacturing process was later standardized and mechanized. After preparation, the dry PermeaPad barrier is currently laser-cut into small disks, i.e., pads, which are ready-to-use barriers. suited for permeability studies in Franz cells [66] and side-by-side cells [67] (Figure 5a,c). Permeapad is also currently available in a 96-well plate ready-to-use format (Figure 5b).

As the major constituents of this barrier are immobilized phospholipid vesicles, PermeaPad could be seen as a direct descendent of the phospholipid vesicle-based permeation assay (PVPA), introduced by the group of Professor Brandl at UIT-The Arctic University of Norway [68,69,70,71] and further developed by the team of Professor Flaten [72,73,74,75]. The PVPA is prepared by depositing liposomes prepared via film hydration/extrusion on a filter support by means of centrifugation. In contrast to its predecessor, the PermeaPad does not contain preformed liposomes. Instead, in the PermeaPad barrier, a liposomal gel [76] forms upon contact with water (Figure 10). Specifically, the thin, dry phospholipid layer hydrates and swells between the two regenerated cellulose layers (with an estimated cut-off size of 7–10 kDa). The preparation of liposomes by means of thin film hydration/extrusion is time-consuming and difficult to scale up, which has limited the commercial development of the PVPA. The avoidance of the liposome preparation step was one of the keys to the successful commercial development of the PermeaPad.

The original PermeaPad was validated by measuring the permeability of 13 chemically diverse compounds using a Franz cell set-up [66]. The validation process showed that the P_app_ values measured with the PermeaPad barrier exhibited a promising correlation with the P_app_ values measured with both PAMPA and Caco-2 cells, as well as a relatively low standard deviation, i.e., below 10%.

In 2016, the German company innoME GmbH, now Phabioc GmbH, started producing and commercializing PermeaPad barriers. For this purpose, they optimized and scaled-up the manufacturing procedure of the barrier. Furthermore, they designed and prototyped a ready-to-use 96-well plate PermeaPad format, which has been commercially available since 2019. In contrast to conventional 96-well filter insert plates, the barriers are attached at a fixed angle to the PermeaPad 96-well insert plate, as depicted in Figure 5b-II. Through the use of this angled position, the formation of air bubbles below the barrier is avoided. In conventional filter insert plates (Figure 5b-I), air bubbles are easily trapped when the two 96-well plates are combined, which may bias the experimental outcome.

For the validation of the commercial version, Jacobsen et al. measured the permeability of 15 structurally diverse compounds using the PermeaPad plate under pH gradient conditions, i.e., the donor solution at pH 6.5 and the acceptor at 7.4, respectively [77]. They demonstrated a good correlation between the PermeaPad plate P_app_ results and the Caco-2 permeability dataset. Furthermore, they showed that the correlation between the PermeaPad plate P_app_ and the human fraction absorbed (A%) resembled the ideal situation, as depicted in Figure 4c, thus validating the IVIVC of this model. In another part of this study, Jacobsen et al. investigated the structure of the PermeaPad barrier via light microscopy [77]. This resulted in the first pictures of the peculiar liposomal structures that formed between the two cellulose sheets upon hydration, i.e., a constrained vesicular phospholipid gel [76].

The interspaces between the phospholipid vesicles are considered to resemble the paracellular spaces found in cell monolayers (Figure 10). Small, hydrophilic molecules can therefore permeate through this para-liposomal space, similarly to what happens in biological tissues [78]. Eriksen et al. showed that the size of this para-liposomal space could be modulated by osmotic-stress-induced liposome shrinking, which increased the permeability of the hydrophilic compound calcein but not the lipophilic compound celecoxib [78]. Even though PermeaPad can mimic paracellular-like transport, it should be mentioned that paracellular permeation enhancers cannot be studied in PermeaPad since these act mainly on proteins of the tight junctions [79].

To verify the stability and compatibility of the PermeaPad barrier with excipients and in vivo-like conditions, Bibi et al. studied the permeability of the hydrophilic marker calcein using Franz cells and side-by-side cells in the presence of various solvents, surfactants, and fast/fed-state simulated intestinal fluids, i.e., FaSSIF and FeSSIF [80]. The PermeaPad barrier demonstrated the ability to sustain different solvents, e.g., ethanol up to 4% and DMSO up to 10%, and aggressive surfactants (including Triton X, which is known to damage both PVPA barriers and Caco-2 monolayers), and fasted/fed-state simulated intestinal fluids. The high stability of the PermeaPad barrier is likely related to its sandwich design, in which the direct contact of phospholipids with the medium is avoided. This design has, until recently, somehow limited the flexibility of PermeaPad’s lipid composition. In contrast to PAMPA, in which the lipid composition can be easily altered, PermeaPad utilizes only a single simple lipid composition. However, in 2022, a PermeaPad skin-mimicking barrier with a varied lipid composition resembling that of the stratum corneum, was successfully validated [81] and will soon enter the market (expected launch: 2023).

Similarly to PAMPA, Permeapad has been used to study the permeability of drugs in the presence of solubilizing excipients such as cyclodextrins [82,83,84,85] and poloxamers [86,87]. Due to its extremely high mechanical and chemical stability, PermeaPad has also found applicability in enabling formulation testing, including the testing of lipid-based formulations such as self-nanoemulsifying drug delivery systems (SNEDDSs) [88], which can contain very high amounts of surfactants, and phospholipid-based dispersions [67,89]. Lipid digestion is an important parameter related to the in vivo performance of lipid-based formulations. Therefore, Bibi et al. combined in vitro lipolysis and permeation in one simultaneous in vitro model based on the use of side-by-side cells with PermeaPad as the barrier to investigate SNEDDSs containing cinnarizine [90]. For this combined model, barrier stability in the presence of the SNEDDS excipients and pancreatic extract was essential. PermeaPad was found to be stable in the presence of the pancreatic extract and SNEDDS excipients, i.e., Cremophor RH 40 (45% W/W), oleic acid (15.4%), Brij 97 (9%), sesame oil (20.6%), and ethanol (10%). The most important outcome of this study was the observation that cinnarizine in SNEDDS showed significantly higher permeability under lipolytic conditions, showing a positive digestive effect, as occurred in vivo. Later, Klitgaard et al. used a similar lipolysis/permeation approach to investigate four different SNEDDSs containing cinnarizine and to test whether such a lipolysis/permeation model could predict the in vivo performance of those formulations (see details below) [88].

Testing enabling formulations introduces two additional levels of complexity into the experimental data analysis process. First, in order to measure the P_app_ through the barrier, it is essential to know/estimate the concentration of the free drug in the donor compartment, c_f_ (Figure 11), which is the amount of drug not associated with nanocarriers, e.g., micelles. The free fraction of the drug is the real driving force for permeation. In fact, according to the theory on dissolution/permeation that is currently most widely accepted [91,92], when enabling formulations are tested, Equation (6), which was described in Section 3.1, should be written as:(13)$$P_{\mathrm{app}}=\frac{j}{c_{f}}$$

Second, when testing enabling formulations, the mass transfer profile over time (Figure 11b) may not be linear. As the free drug fraction is defined by the nanocarrier-drug equilibrium constant, K_eq_ (Figure 11a) [93,94] and c_f_. As a consequence, the effective gradient (Figure 11a) is likely to decrease over time. In this case, the performance of a formulation can be evaluated on the basis of the in vitro AUC, i.e., the area under the curve of the in vitro mass transfer profile, i.e., the amount multiplied by time (Figure 11c). This data interpretation approach is very efficient when it comes to enabling formulation testing of any kind in cases where it is difficult to quantify c_f_. During their evaluation of cinnarizine SNEDDSs, Klitgaard et al. [88] cleverly applied the concept of the in vitro AUC, which was previously applied in an artificial gastrointestinal tract model simulating human digestion. With this approach, they showed an almost linear correlation (R^2^ = 0.92) within in vivo and in vitro data, proving the suitability of the lipolysis/permeation model with the use of PermeaPad as a barrier for the screening of lipid-based formulations [88].

Using different permeation devices and testing different enabling formulations, the capability of the PermeaPad barrier to predict in vivo performance has been demonstrated for various phospholipid-based dispersions of celecoxib [88], supersaturating formulations of posaconazole [95], and enabling formulations of dipyridamole [96]. On the other hand, two studies have provided evidence of a lower IVIVC for PermeaPad [97,98].

Even though PermeaPad was developed for studying oral drug absorption in vitro, the barrier has also been successfully employed to study enabling formulations for buccal [99] and nasal [100] administration. In a recent study on enabling formulations for nasal administration, an artificial mucus layer was successfully added on top of the PermeaPad barrier to better mimic the nasal mucosal environment [101]. Due to its design, PermeaPad might not be suited for the testing of macromolecules with a molecular weight larger than 10–15 kDa.

## 5. Lipid-Free Membranes for In Vitro Permeability/Absorption Profiling

In this section, we briefly discuss lipid-free, commercially available membranes that have been used to study drug permeability/absorption. We here use the term membrane to indicate a thin sheet (or layer of sheets) which consists of fibrous material.

As discussed above, for a barrier to be biomimetic, its resistivity to permeation must account for both solute partitioning and diffusivity (Equation (11)) as in vivo biological barriers. In the models discussed above, i.e., PAMPA and PermeaPad, partitioning is governed by the presence of lipids and, in the case of PAMPA, organic solvents and lipids. However, in a thin sheet of fibrous material such as regenerated cellulose (schematized in Figure 12b), the drug will diffuse through the hydrated pores with negligible partitioning. It follows that drug permeability profiling performed using such thin lipid-free membranes will be greatly influenced by the solute’s diffusivity in water.

In fact, according to the Stokes–Einstein equation (Equation (14)), and assuming full hydration of the membrane, drug permeation is controlled mostly/solely by the drug’s diffusivity in water, D_w_, and therefore by its hydrodynamic radius, r, as follows:(14)$$D_{w}=\left(\frac{k_{b}T}{\eta \times r\times 6\pi }\right)$$
where η is the medium viscosity, T is the absolute temperature, and k_b_ is the Boltzmann constant. It is our opinion that regenerated cellulose, which is the most commonly used lipid-free membrane in flux studies applications, should therefore be considered a “size-exclusion” membrane rather than a biomimetic barrier, as the overall permeability measured through regenerated cellulose is greatly, if not solely, controlled by the molecule’s shape and size (Equation (14)). It should be mentioned that lipid-free membranes composed of more lipophilic materials, as compared to regenerated cellulose, can account for partitioning to some extent. One example is polydimethylsiloxane (PDMS) membranes [102,103,104]. PDMS membranes can be prepared by casting using the commercially available Sylgard 184 silicone elastomer kit [102] or they can be obtained as ready-to-use PDMS sheets, e.g., Silatos silicone sheeting [105].

### 5.1. Regenerated Cellulose

As mentioned above, regenerated cellulose (schematized in Figure 12b) with different molecular weight cut-offs has been used for in vitro drug permeability screening. In contrast with porous materials, these sheets are not characterized by the presence of micrometric pores of homogeneous size (Figure 12a) but rather by a tight, non-homogeneous net of cellulose fibers (Figure 12b). The higher the density of the fibers, the smaller the molecular weight cut-off size. This material is commonly employed for separation, e.g., ultrafiltration [106] and dialysis purposes. Di Cagno et al. showed that regenerated cellulose was not suitable for the permeability profiling of drugs due to its lack of selectivity [66]. In fact, as shown in in Figure 13, the P_app_ values obtained with regenerated cellulose were generally high in comparison to those of other in vitro permeability assays (≤2 × 10^−5^ cm/s) and correlated well with the compounds’ molecular sizes (in Figure 13 we show molecular weight for simplicity). Berben et al. introduced the artificial membrane insert system, AMI, a very simple set-up based on two chambers, i.e., donor and acceptor, separated by a regenerated cellulose sheet with a small molecular weight cut-off of 2 kDa [64]. They measured the P_app_ values of 14 drugs in the AMI system, obtaining a satisfactory correlation with the Caco-2 model. A closer analysis of the data, however, raises suspicions regarding the strong correlation between P_app_ and molecular size, similarly to what has been observed in other studies [66].

Regenerated cellulose has been proposed as a barrier for advanced dissolution/permeation testing of enabling formulations, especially amorphous solid dispersions (ASDs). Sironi et al. successfully employed regenerated cellulose membranes on a flow-through diffusion cell system named PermeaLoop (system not commercially available) to study the dissolution/permeation behavior of an ASD of linifanib, a BCS class II compound [107]. The same group later employed this approach to study different supersaturating formulations of posaconazole (BCS class II) [95]. This set-up, which combined the PermeaLoop with regenerated cellulose, enabled them to better capture the initial short period of supersaturation of the acidified posaconazole suspension, as compared to the μFLUX system employing the PAMPA barrier (μFLUX is discussed below in Section 6). Wilson et al. evaluated the performance of a series of enzalutamide (BCS class II) ASDs in vitro and in vivo. For the in vitro evaluation, they studied enzalutamide mass transfer rates across a regenerated cellulose membrane in side-by-side cells. The mass transfer rates correlated well with enzalutamide absorption in rats [108]. Using a 96-well plate system consisting of regenerated cellulose, i.e., a PermeaPlain plate from InnoME GmbH, Jacobsen et al. evaluated the performance of crystalline and amorphous tadalafil (BCS class II) formulations, including a Soluplus ASD. In this case, the in vitro study also correctly ranked the formulations in terms of their in vivo absorption in rats [109].

On the other hand, regenerated cellulose seems less suited to the study of lipid-based enabling formulations such as liposomes. Wu et al. tested different liposomal formulations containing moderately permeable hydrocortisone using regenerated cellulose membranes and PermeaPad in Franz cells [93]. The results obtained using the regenerated cellulose were totally unpredictive in the ex vivo situation [110], overestimating the drug permeability of hydrocortisone two-fold. In contrast, the results obtained with the PermeaPad coincided perfectly with previous results obtained ex vivo utilizing sheep nasal mucosa [110].

Recently, Braeckmans et al. investigated the mechanism behind the pronounced positive food effect of abiraterone (BCS class IV) [111]. For this purpose, they conducted both in vitro permeability experiments in the AMI system, i.e., using a regenerated cellulose barrier, in the presence of different biorelevant media and lipids, as well as in situ rat perfusion studies. In this study, the permeability measured with the AMI system was not predictive for the outcome of the in situ rat perfusion studies. It should be mentioned that the donor medium used for the AMI study and the in situ rat perfusion where not identical, i.e., drug supersaturation vs. solution.

Another issue with regenerated cellulose membranes is that due to their lower and unselective resistance to permeation, this type of barriers cannot be employed in permeability studies with a pH/tonicity gradient [66].

To conclude this section, the applicability of regenerated cellulose for in vitro permeation/absorption studies should be carefully considered. As discussed above, regenerated cellulose is not suitable for drug permeability profiling.

### 5.2. Strat-M^®^

Thus far, the main area of application of the in vitro permeability assays discussed in this paper has been oral drug delivery. However, it is worth mentioning one commercial cell free model that was developed for studying drug penetration through the skin—Strat-M^®^ (Strat-M, Merck Millipore, Danvers, MA, USA). Similarly to the previously discussed barriers, Strat-M is also used to study the passive diffusion of drugs and it is commercially available. This technology was developed upon the observation that the resistivity to penetration between the stratum corneum and some types of materials, such as silicone membranes [112], was to some extent comparable. The Strat-M membrane has an asymmetrical lipid-free three-layer structure that was engineered to mimic the structure of healthy human skin, but not its composition. The three layers differ in their porosity and thickness in order to mimic the different layers of human skin, i.e., the epidermis, dermis, and subcutaneous tissue. According to the available Merck technical note [113], Strat-M is composed of two layers of polyethersulfone (PES), which is highly resistant to solute diffusion, on top of one layer of polyolefin, which exhibits a lower resistivity than PES to solute diffusion. These layers create a porous structure with a gradient across the membrane in terms of pore size and diffusivity. Strat-M is currently available in pre-sized round disks of different sizes (up to 25 mm in diameter) to be used in classic diffusion cells (Figure 5a,c) and is therefore not very well suited for high-throughput drug/formulation screening. Haq et al. recently demonstrated high similarity between human skin and Strat-M in regard to the permeability of nicotine [114]. They also proved that Strat-M was a reliable tool for in vitro formulation screening for the topical administration of nicotine, demonstrating high similarity to human skin. Recently, Arce et al. demonstrated a good correlation within drug penetration through Strat-M and porcine skin, i.e., the gold standard in skin studies [115].

## 6. Commercially Available In Vitro Dissolution/Permeation Systems

As stated above, the only in vitro test that is accepted for biowaivers by the FDA and EMA is the comparative dissolution test, which reaches its predictive limit when testing formulations of poorly soluble drugs, as well as formulations that contain excipients that may influence oral absorption and in biorelevant media, e.g., bile salts. Due to the technological advancements in the field of cell-free permeability models, several dissolution/permeation systems have been proposed. Those systems mimic the in vivo scenario, and therefore they hold the potential to expand the BCS-based biowaiver approach to drugs and formulations for which biowaivers cannot currently be granted. In the literature, a multitude of dissolution/permeation systems have been described that differ in terms of their size, geometry, agitation method, barrier type, etc. An overview of those systems has been provided by O’Shea et al. [116]. In this section, we review the dissolution/permeation systems based on biomimetic barriers that are commercially available.

Combined dissolution/permeation systems are specifically designed for studying the impact of formulations, mainly solid dosage forms, on oral drug absorption. One of the most successful dissolution-permeation systems is µFLUX from pION Inc. (Billerica, MA, USA) [117]. This system is a semi-automatized horizontal diffusion system, i.e., side-by-side diffusion cells (Figure 5c), in which the acceptor and donor chambers are separated by a PAMPA barrier. µFLUX is equipped with state-of-the-art UV detectors in both the acceptor and the donor compartment in order to simultaneously obtain dissolution vs time (Figure 14b) and a mass flux vs time (Figure 14c) profiles. This system has found wide implementation in both academia and industry [118,119,120,121], even though the small size and shape of µFLUX might be a disadvantage when testing complete solid dosage forms.

The PERMETRO system from Logan Instruments Corp. is a semi automatized vertical diffusion cell, conceptually similar to the µFLUX, that currently utilizes the PermeaPad. In this system, samples are withdrawn automatically from the acceptor compartment at defined time points and placed in vials for quantification [81].

Other commercially available dissolution/permeation systems are macroFLUX and bioFLUX (Pion Inc., Billerica, MA, USA). This system (schematized in Figure 14a) consists of a traditional dissolution vessel (ranging from 250 mL up to 900 mL) with a paddle stirrer. In this case, the biomimetic barrier PAMPA is mounted on a special insert that contains the acceptor medium. This system allows for real-time quantification of the drug concentration in both the donor, i.e., the dissolution vessels, and the acceptor via UV fiber optic detectors. The bioFLUX system has demonstrated high in vivo predictability for enabling formulations of BCS class IV drugs, e.g., itraconazole [122].

The Dissoflux system (Electrolab India PVT. LTD, Mumbai, India) resembles macroFLUX, but until now this system has been mostly equipped with PermeaPad barriers. This system consists of eight standard 900 mL dissolution vessels, each of which can accommodate a PermeaPad cartridge, i.e., the acceptor (max volume: 10 mL, permeation area: 3.14 cm^2^). This system currently finds industrial applications in the generic drug product development. As an example, Figure 15 shows the dissolution/permeation profiles obtained with the use of Dissoflux equipped with PermeaPad cartridges (unpublished data obtained from Electrolab India PVT. LTD). In this example, an approved marketed solid dosage form and a new generic formulation of a BCS class IV drug were compared. When using the in vitro AUC_0–6 h_ obtained from the permeation profile (gray area in Figure 15) as the parameter for comparison, bioequivalence between the two formulations was proven, which was confirmed in vivo. In contrast, the dissolution profile was not conclusive (Figure 15, small box). This is an interesting example highlighting the potential of dissolution/permeation systems to be used for biowaivers of poorly soluble compounds such as BCS class II and IV drugs.

## 7. Conclusions

In this review, we have summarized the state of the art of commercially available in vitro cell-free permeation tests which, if properly employed, have the potential to boost sustainability in pharmaceutical development by refining, reducing, and replacing in vivo studies. In this review, we focused mostly on tools for studying oral drug delivery, the administration route for which a BCS-based biowaiver legislative framework has been recently introduced. However, skin and mucosal drug delivery have also been discussed to some extent.

These tools have exhibited significant advantages in respect to cell-based assays, and their conscious implementation can lead to correct and cost-/time-effective predictions in early-stage drug development, with limited need for animal testing. When necessary, for better comprehension of the in vivo scenario, it might be useful to combine different and complementary permeation models, such as cellular models that are able to quantify carrier-mediated transport and transcytosis.

In this review, we also presented commercially available dissolution/permeation set-ups equipped with cell-free permeation barriers. We believe that those approaches are promising as they could be exploited for expanding the BCS-based biowaiver approach to all classes of compounds, i.e., from BCS class I compounds to class IV compounds. Nevertheless, the full-scale implementation of such in vitro methods by the pharmaceutical industry requires a political will and new and modern legislative frameworks regulating drug development.

## Figures and Tables

**Figure 1 pharmaceutics-15-00592-f001:**
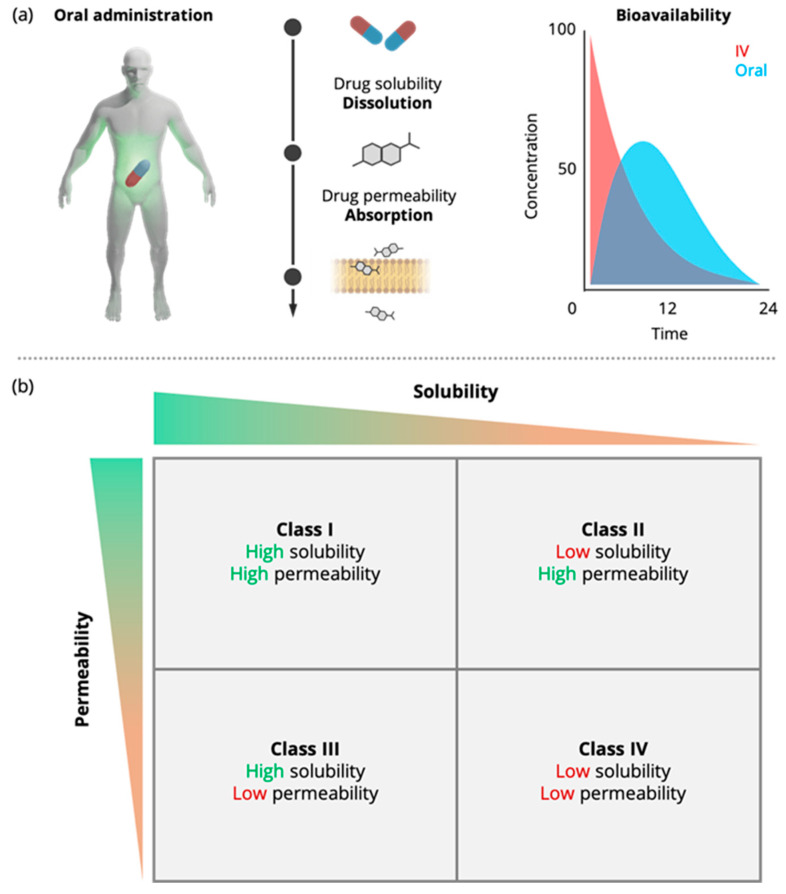
Schematic representation of the interplay of solubility and permeability in oral drug absorption (**a**) and the Biopharmaceutics Classification System (BCS) introduced by Amidon et al. in 1995 [5] (**b**). Once a dosage form is administered orally, it undergoes dissolution and drug molecules are then absorbed through the gastrointestinal (GI) tract through the process of permeation. The fraction dose absorbed, %A, depends mainly on the drug’s aqueous solubility and its permeability through biomimetic barriers.

**Figure 2 pharmaceutics-15-00592-f002:**
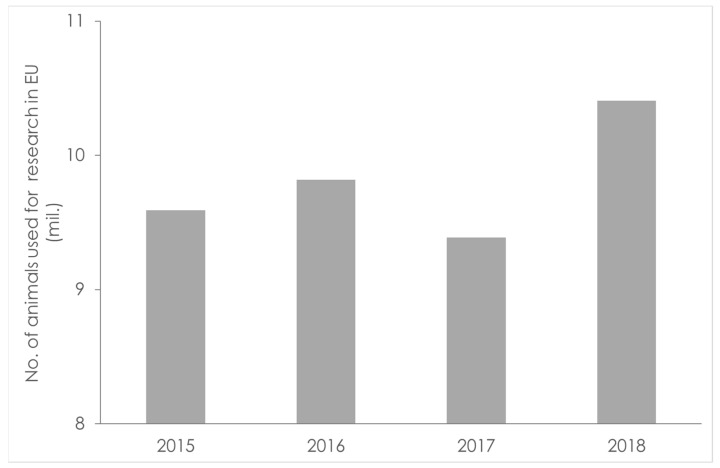
The numbers of animals used for research and testing in the EU in the period 2015–2018 [11].

**Figure 3 pharmaceutics-15-00592-f003:**
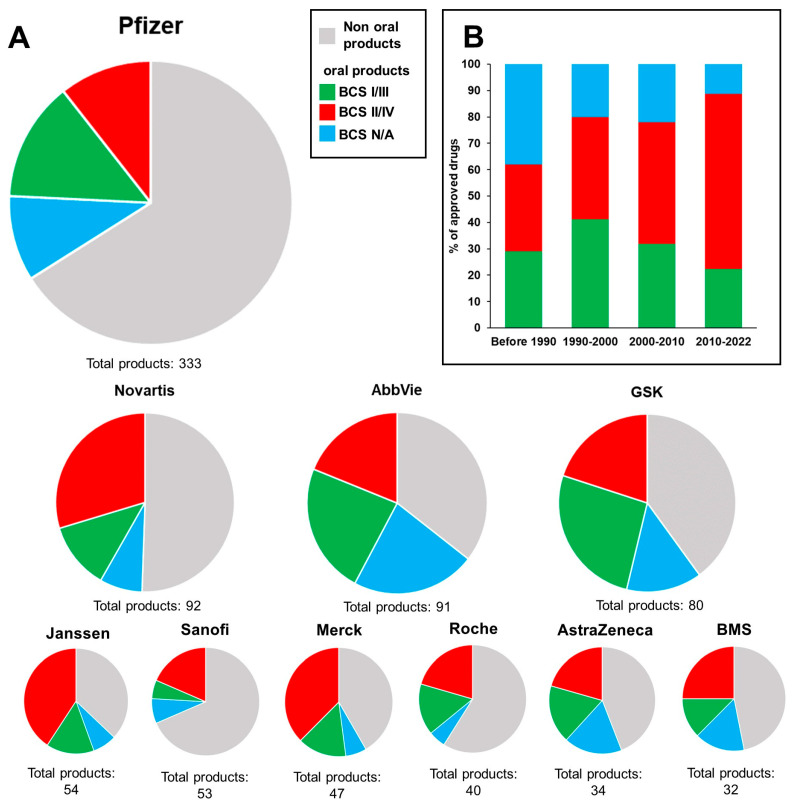
Overview of drug products from the 10 largest pharmaceutical companies (Table 1). (**A**) Drug product portfolios, in which products are divided into non-oral and oral products. Oral products are divided into high-solubility (BCS class I/III) and low-solubility (BCS II/IV) drugs. (**B**) BCS classes of oral drugs from the 10 pharmaceutical companies by their year of approval. BCS class I/III—green; BCS class II/IV—red; BCS class not available (N/A)—blue.

**Figure 4 pharmaceutics-15-00592-f004:**
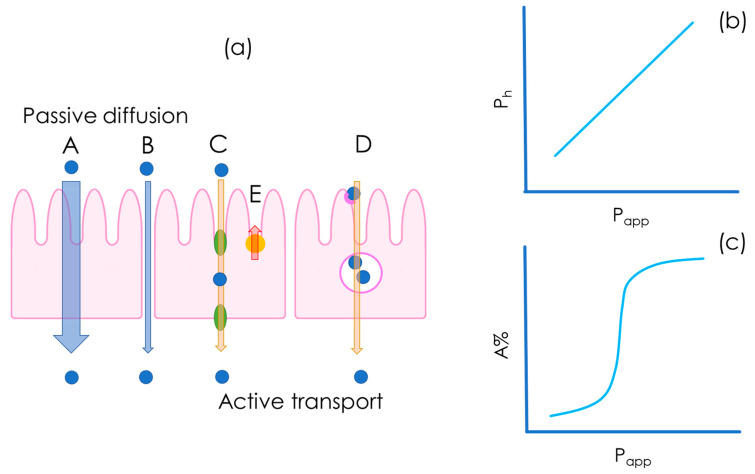
Pathways for drug absorption across an epithelial monolayer (**a**), the expected correlation between in vitro apparent permeability (P_app_) and in vivo human permeability (P_h_) (**b**), and the expected correlation between P_app_ and the fraction absorbed (A%) (**c**). Drug absorption can occur via the transcellular route (**A**), the paracellular route (**B**), carrier-mediated transport (**C**), or transcytosis (**D**). Efflux transporters actively transport the drug in the opposite direction (basolateral–apical, (**E**)).

**Figure 5 pharmaceutics-15-00592-f005:**
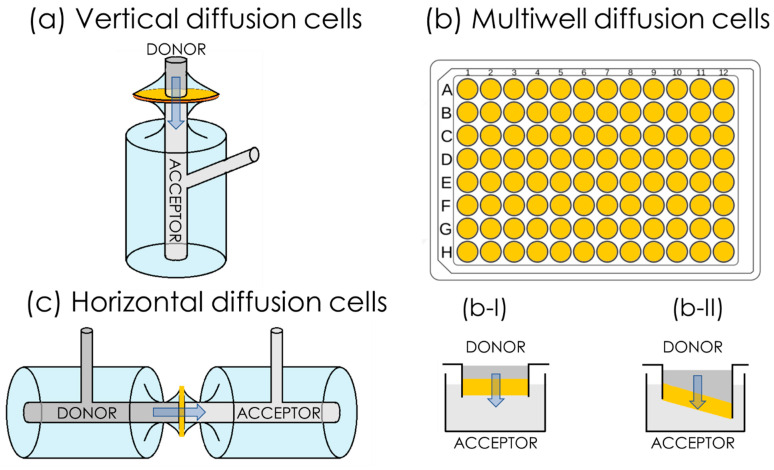
Schematic representation of set-ups for permeability assays. Permeability assays are based on two compartments, the donor and the acceptor, separated by a (biomimetic) barrier, i.e., the yellow segments in the figure. In general, permeability tests can be performed utilizing macroscopic diffusion cells, in which the diffusion of the drug takes place vertically, i.e., from top to bottom (or the reverse) (**a**), or horizontally, i.e., from side to side (**c**). In order to increase the potential of this method for high-throughput screening (HTS), several permeability assays are available in the multiwell-plate format (**b**), in which a (biomimetic) barrier is attached to the top plate horizontally (**b-I**) or at an angle (**b-II**) in order to reduce the risk of air bubble formation.

**Figure 6 pharmaceutics-15-00592-f006:**
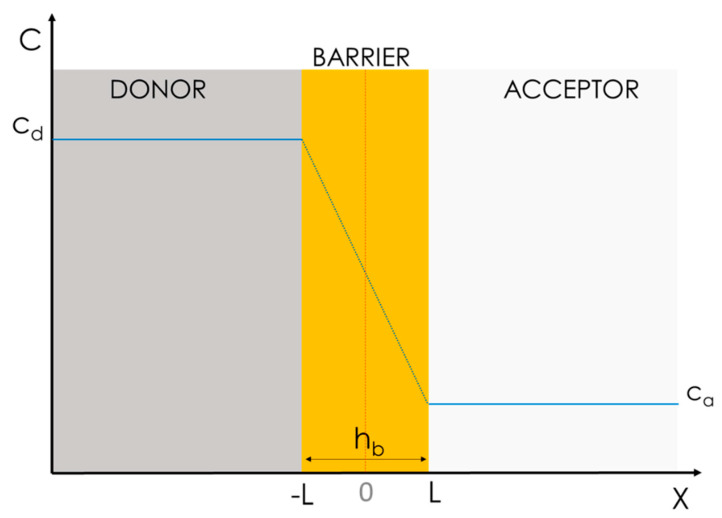
Representation of a conventional physical model used to interpret the passive transport of a compound through a barrier of thickness h_b_. The blue lines represent the changes in the concentration gradient within the donor (c_d_) and acceptor (c_a_). In this model, there are no boundary regions, i.e., interfaces, within the water phases of the donor and acceptor media and the barrier (X = −L and X = L, respectively).

**Figure 7 pharmaceutics-15-00592-f007:**
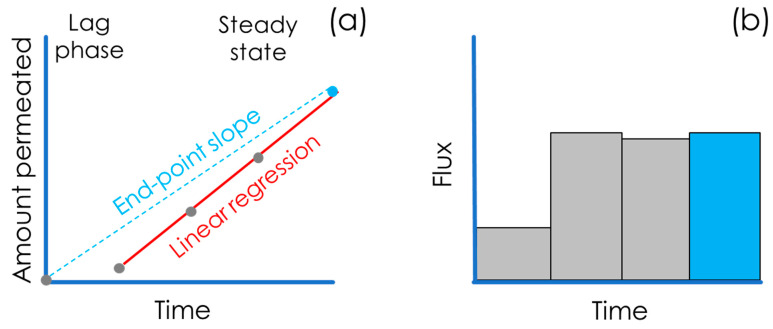
Plot of the amount of drug permeated vs time (**a**) and profile of the flux vs time (**b**). In general, permeability assays suffer from a lag phase, in which the flux is not constant. The interpolation interval of the linear regression ((**a**) red line) for measuring the mass flux should thus be carefully selected in order to avoid the misinterpretation of the data and achieve the best possible coefficient of determination (e.g., >0.9). The end-point approach for calculating P_app_ ((**a**) light-blue line) leads to lower absolute P_app_ values than in the multiple-point approach.

**Figure 8 pharmaceutics-15-00592-f008:**
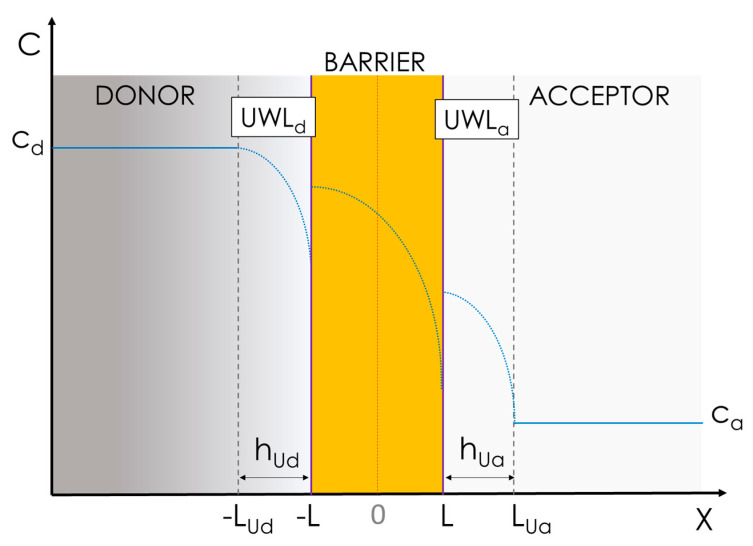
Advanced physical model to describe the permeation process in vitro. This model considers the existence of unstirred water layers in the donor and acceptor, namely, UWL_d_ and UWL_a_, as additive thin layers that need to be crossed by the solute. Moreover, in this model, the different phases (water–lipid–water) are considered to be separated by boundary regions (interfaces, at X = −L and X = L).

**Figure 9 pharmaceutics-15-00592-f009:**
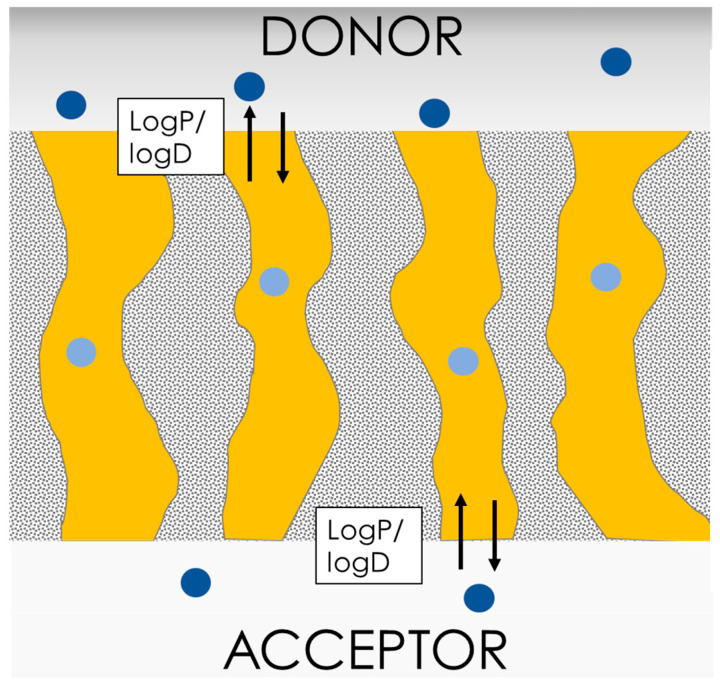
Schematic representation of the PAMPA barrier structure. The PAMPA barrier is a porous filter (0.22–0.45 µm pore size) that is soaked with a lipid mixture (yellow section). The rate-limiting step, controlling permeability, in this case consists mainly of the solute partition/distribution coefficient (LogP/logD) and the diffusivity in the lipid barrier (D_B_). In the case of electrolytes, the molecules need to be neutralized (light blue dots) and then re-gain their charge once in the acceptor.

**Figure 10 pharmaceutics-15-00592-f010:**
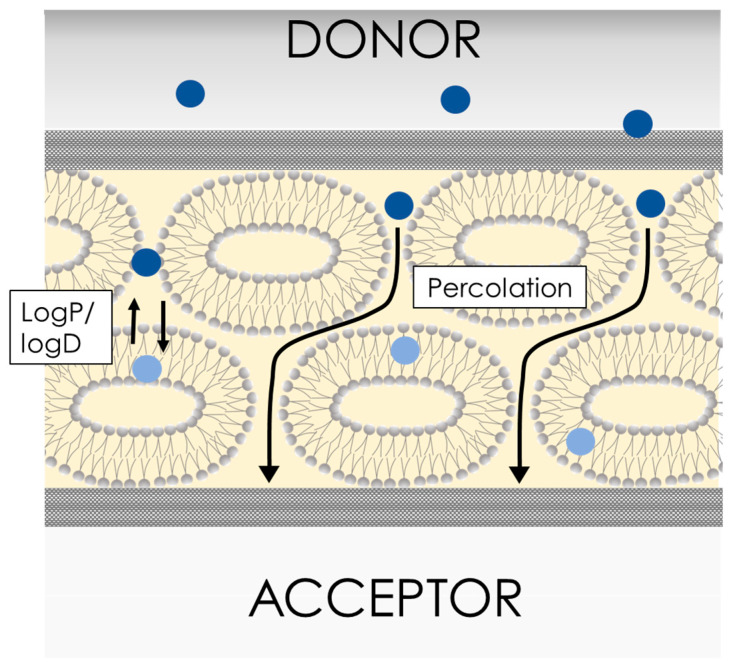
Schematic representation of a hydrated PermeaPad. This barrier is constituted by a liposomal gel contained within two low-retention inert layers. The drug molecules can diffuse through the liposome bilayers, i.e., exploiting partitioning, and/or they can diffuse directly through the para-liposomal space.

**Figure 11 pharmaceutics-15-00592-f011:**
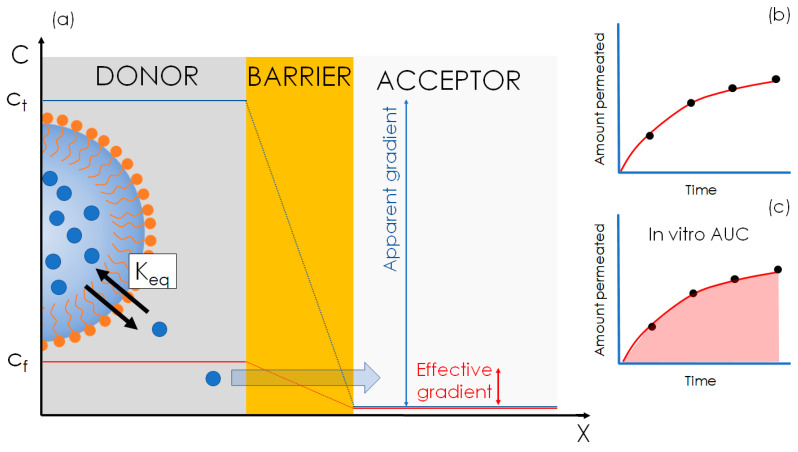
Simplified permeability model for a nanocarrier-based enabling formulation (**a**) and the corresponding mass transfer profile (**b**) and the in vitro AUC (**c**). According to the free-fraction theory [93], the effective gradient for permeation should be given by the free drug fraction, c_f_, i.e., the unloaded drug. The mass flux profile in this case is highly influenced by the equilibrium constant (K_eq_) of the loaded (in the nanocarrier) to the unloaded (free) drugs.

**Figure 12 pharmaceutics-15-00592-f012:**
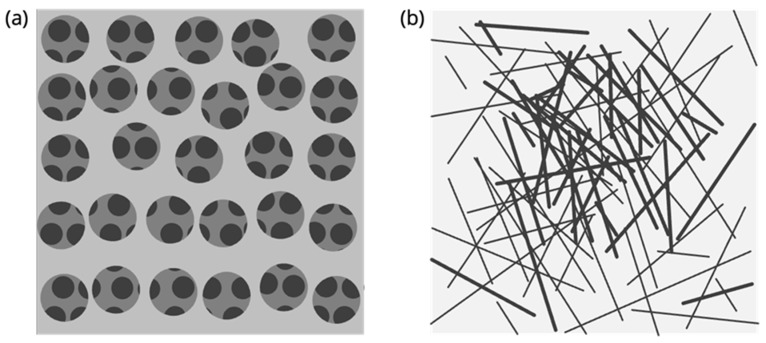
Structural differences between a porous material (**a**) and a fibrous material (**b**).

**Figure 13 pharmaceutics-15-00592-f013:**
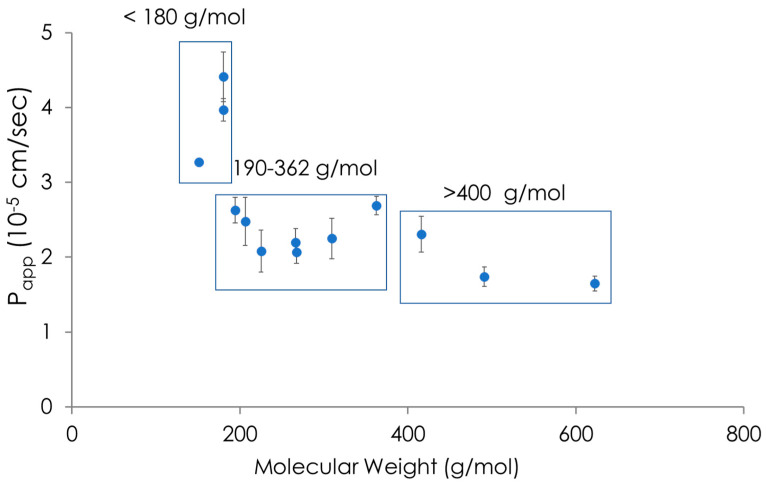
Correlations of P_app_ values measured using regenerated cellulose membranes and molecular weights (original data form Di Cagno et al. [66]). A quite evident correlation of P_app_ ≈ 1/mw emerged.

**Figure 14 pharmaceutics-15-00592-f014:**
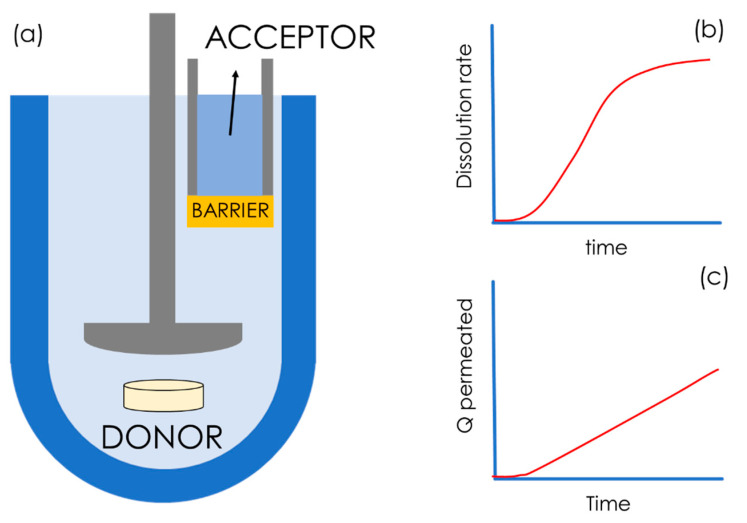
Schematic representation of a dissolution-absorption apparatus (**a**), e.g., macroFLUX, bioFLUX, and Dissoflux. Solid dosage forms are placed in the donor compartment, i.e., a dissolution vessel, under paddle stirring. The drug concentration is monitored in the donor and acceptor compartments, allowing the user to create dissolution vs time profiles (**b**) and drug permeated vs time profiles (**c**), respectively. In such set-ups, high mechanical and chemical stability of the barrier over time are essential.

**Figure 15 pharmaceutics-15-00592-f015:**
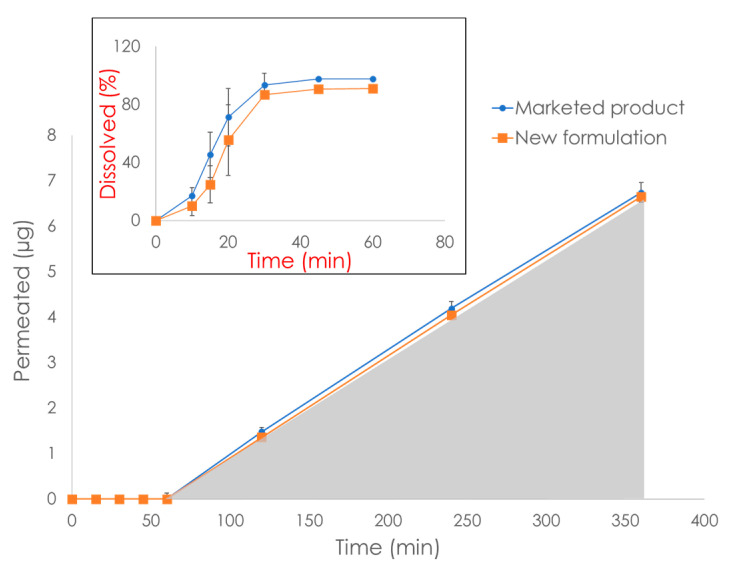
Example of dissolution/permeation results obtained with Dissoflux with PermeaPad cartridges. The in vitro AUC_0–6 h_ (gray zone) was used to evaluate the bioequivalence of two solid dosage forms (marketed product and new formulation, personal communication from Electrolab). The dissolution media consisted of 0.05 M sodium phosphate (pH 6.8) and 1% tween 80.

**Table 1 pharmaceutics-15-00592-t001:** Publication metrics on the topic of “in vitro permeability testing” for the 11 largest pharmaceutical companies worldwide.

Company Name	Location of Headquarters	Revenue 2021 (Bill. USD) [18]	No. Results	Articles Published by Affiliation (%)
Pfizer Inc.	New York, USA	81.3	232	0.4
Sinopharm	Shanghai, CN	60.5	1	0.1
AbbVie	Chicago, USA	56.2	58	1.4
Johnson & Johnson	New Brunswick, USA	52.1	50	0.9
Novartis	Basel, CH	51.6	92	0.4
F. Hoffmann–La Roche SA	Basel, CH	49.3	74	0.5
Merck & Co. Inc.	Kenilworth, USA	48.7	149	0.3
GlaxoSmithKline	Brentford, UK	47.9	114	0.4
Bristol Myers Squibb	New York, USA	46.4	96	0.6
Sanofi S.A.	Paris, FR	44.6	61	0.3
AstraZeneca	Cambridge, UK	37.4	144	0.7

Source: Scopus, last accessed August 2022.

## Data Availability

A list of drug products from 10 large pharmaceutical companies, on which Figure 3 is based, is available in the Appendix A.

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
