# Peer review of "Commercially Available Cell-Free Permeability Tests for Industrial Drug Development: Increased Sustainability through Reduction of In Vivo Studies"

_pharmaceutics, 2023, doi:10.3390/pharmaceutics15020592_

Round 1

Reviewer 1 Report

Title: Commercially available cell-free permeability tests for industrial drug development: Increased sustainability through reduction of in vivo studies

The review addresses the issue of drug development which generally relies on clinical and preclinical studies and seeks to highlight the value of using cell-free in vitro testing systems for the testing and evaluation of drugs. The general topic is one of high interest. However, the review fails to address the value and limitations of the use of in vitro models in the drug development process objectively and comprehensively. Overall, the paper lacks cohesiveness, a comprehensive approach to the parameters that need to be considered in drug development studies and fails to provide a compelling picture of the state of the in vitro drug testing field.

Major issues:

·      The overall framing of the study is not cohesive – it mentions sustainability, carbon footprint, limitations of in vivo human studies, in vivo testing but without a clear focus and connectivity

·      The authors mention that “Unfortunately, over the years, some theoretical misconceptions linked to these assays have somehow taken root among a part of the scientific community” but they do not directly mention or address these misconceptions, which would greatly increase the value of the review paper

·      The literature survey approach described may not be the most adequate as many pharmaceutical companies do not publish their work due to intellectual property considerations

·      Section 2 does not bring much value to the review; pharmaceutical companies typically use standardized, validated tests for regulatory approval, therefore a search of validated in vitro tests or a patent search could have been more pertinent for this review

·      There is no discussion on the effect of drug molecule MW and/or LogP on the passive diffusion through PAMPA other than a brief mention of LogP/LogD in Figure 9; such a discussion would be particularly pertinent in the context of the equivalence of the lipid barriers used to physiological membranes

Minor issues:

·      Remove period in the title

·      Review entire manuscript for English syntax correctness

·      Suggest describing in more detail what is a biowaiver

·      The scope of Figure 1a is confusing as is and it does not clearly convey the relationship between drug solubility and permeability in oral drug absorption

·      Define abbreviations in figure captions (e.g., HTS in Figure 5 caption)

·      There is no mention of the validation of PermaPads with available in vivo data, only with other in vitro testing systems

Author Response

The authors are grateful for the valuable and insightful comments from the expert reviewer and for the time and effort, which helped to improve this comprehensive review. We have addressed all objective points raised by the reviewer. Alterations to the original manuscript text have been highlighted in yellow. We also took the chance to proof read the manuscript. Eventual alterations to the original text are highlighted in grey

The review addresses the issue of drug development which generally relies on clinical and preclinical studies and seeks to highlight the value of using cell-free in vitro testing systems for the testing and evaluation of drugs. The general topic is one of high interest. However, the review fails to address the value and limitations of the use of in vitro models in the drug development process objectively and comprehensively. Overall, the paper lacks cohesiveness, a comprehensive approach to the parameters that need to be considered in drug development studies and fails to provide a compelling picture of the state of the in vitro drug testing field.

We thank the reviewer for this comment. Indeed, it is a very comprehensive review making it difficult to achieve cohesiveness throughout the whole article. What we tried to achieve with this review was to help any reader, even without a background in biopharmaceutics, to reflect on the issue of the limited implementation of fully artificial in vitro permeability methods, which are advantageous in the context of the 3Rs principle and from a sustainability point-of-view, especially in the introduction. The manuscript on the other hands properly refers to more than 100 highly ranked peer-reviewed papers and some of the most referenced review papers in the field of non-cellular permeability studies. We therefore believe that this article provides a compelling picture of the state of the art of in vitro drug testing as a field.

The overall framing of the study is not cohesive – it mentions sustainability, carbon footprint, limitations of in vivo human studies, in vivo testing but without a clear focus and connectivity

We agree with the reviewer that the article can be somehow subdivided in three mayor subtopics: sustainability issue, in vitro implementation, and in vitro state of the art. But this is somehow the scope of this review: to highlight the issues that comes with animal testing, and to be critical to the current industrial approach, which is objectively insufficient to promote a sustainable transition of the field. Through the later sections, which explain the theoretical background and describe the in vitro methods in detail, we hope that the reader will be inspired to see which possibilities lie within these methods and how these can be used in a good way. We are aware that the stronger limitation from an industrial perspective is the lack of a proper, modern legislative framework. On this regard, we have added a sentence in section 2 (highlighted in yellow) to clarify this point. On the overall framing, we have restructured section 2, following the comment of reviewer 2. In lack of additive constructive suggestions to improve the paper, and since reviewer two indicated to be satisfied with the structure, the authors have decided not to implement additional alterations to the general structure of the review, which we regard to be of sufficient clearness.

The authors mention that “Unfortunately, over the years, some theoretical misconceptions linked to these assays have somehow taken root among a part of the scientific community” but they do not directly mention or address these misconceptions, which would greatly increase the value of the review paper

We apologize for the lack of clearness of this point. We have added a sentence in the ending of section 2 that refers to the later section 3.1.1 and 3.1.2 where we actually describe the incongruence in detail.

The literature survey approach described may not be the most adequate as many pharmaceutical companies do not publish their work due to intellectual property considerations

We agree with the reviewer that the data presented in section 2 might be prone to uncertainty due to confidentiality data not shared by companies. However, we regard this approach to be quite constructive as it gives a general, yet clear, picture of the state of the art of in vitro tools implementation. We have added a paragraph to clarify the uncertainty to which such results are prone to in section 2 (before Table 1), and we have restructured section 2 as a whole.

Section 2 does not bring much value to the review; pharmaceutical companies typically use standardized, validated tests for regulatory approval, therefore a search of validated in vitro tests or a patent search could have been more pertinent for this review

We agree with the author that pharmaceutical companies typically use standardized, validated tests for regulatory approval. This section is used to contextualize and give the broad picture of the industrial implementation of such in vitro tools in biopharmaceutics and to highlight the difficulty pharma industry is facing in light of a lack of a proper legislation. We added a sentence in the end of section 2 to highlight this fact (before Table 1), and we restructured section 2.

There is no discussion on the effect of drug molecule MW and/or LogP on the passive diffusion through PAMPA other than a brief mention of LogP/LogD in Figure 9; such a discussion would be particularly pertinent in the context of the equivalence of the lipid barriers used to physiological membranes

We thank the reviewer for this useful comment. We have amended the text accordingly (ending of section 4.1, see page 26 line 545-547) and we have added a relevant reference highlighting the importance of MW and other molecular features for Papp measured with PAMPA-like barriers.

Remove period in the title

We have removed the period from the title.

Review entire manuscript for English syntax correctness

We have revised and proof read the whole manuscript to improve its readability (changes highlighted in grey)

Suggest describing in more detail what is a biowaiver

We have modified the abstract, and we  have added a sentence in section 1.1. Furthermore, we have added an additional reference to the ICH M9 guideline, where a detailed description of the BCS-based biowaiver approach can be found, in section 1.3 (see page 7 line 137).

The scope of Figure 1a is confusing as is and it does not clearly convey the relationship between drug solubility and permeability in oral drug absorption

In order to make the figure more meaningful we have amended the figure caption to make it more descriptive

Define abbreviations in figure captions (e.g., HTS in Figure 5 caption)

We have defined the abbreviations in the captions, when necessary, e.g., Fig. 5

There is no mention of the validation of PermaPads with available in vivo data, only with other in vitro testing systems.

We would like to thank the reviewer for pointing out that the comparison between PermeaPad in vitro data and in vivo data was not described clearly enough in the original manuscript.

We have inserted a comment in section 4.2 to underline the article where the in vitro permeability (Papp) determined using the PermeaPad plate was compared to the fraction absorbed in human (Jacobsen et al. 2020, DOI: 10.1007/s11095-020-02807-x). The PermeaPad plate Papp showed the expected correlation, which is shown on Figure 4(c), and which is similar to the correlation described for PAMPA (Zhu et al. 2002, DOI: 10.1016/s0223-5234(02)01360-0) and Caco-2 (Artursson and Karlsson 1991, DOI: 10.1016/0006-291X(91)91647-U). This section lies after fig. 10 and it is highlighted in yellow (page 28 line 592-594).

We also highlighted the section where in vitro permeation results were compared to in vivo pharmacokinetic data when testing an enabling formulation (SNEDDSs) using a lipolysis-permeation approach employing the PermeaPad barrier (see page 30 line 635-638 and page 32 line 659-663). Furthermore, we added additional references where enabling formulations were tested using permeation devices employing the Permeapad barrier and where the in vitro permeation results are compared to in vivo data (see page 32 line 664-673).

Reviewer 2 Report

Comments

Date: 10-01-2023

Manuscript ID: Pharmaceutics-2125201

Title: “Commercially available cell-free permeability tests for industrial drug development: Increased sustainability through reduction of in vivo studies”.

Jacobsen et al. addressed various aspects of drug permeation for industrial drug development using in vitro permeation models and tools to waive in vivo animal utility. The review is well designed, well written and informative to readers. I appreciate to present valuable information in the review. The compilation is interesting and informative to reader working in the domains in academics as well as industry. However, I recommend major suggestions before publication.

Comment 1: Abstract must be short and concise. The used abbreviations must be defined appeared first time in the text body such as BCS, ICH, and ADME etc. There must be consistency in using “in vitro” (must be italic or non-italic). Copyright trade mark must be removed (PermeaPad®). In the sentence “……predicting the in vivo situation indicating that…”, define the in vivo situation? Please replace the sentence “generic drugs development” with “generic product development”.   

Comment 2: In the sentence “From 2016 to 2017, this number decreased slightly (-4.4%) [11]. Sadly, the slight decrease was followed by a significant increase by over 10% in 2018”, I found contradictory observation after comparing figure 2. Please check

Comment 3: There must be a separate search methodology section along with inclusion and exclusion criteria.

Comment 4: Page 7, line 211-225, I suggest to include reference source for the content. Table 1 must be corrected for the last column (percent correction required)    

Comment 5: Correct the sentence “the field of pharmaceutics”. Please provide a suitable reference against the sentence “Permeability can be defined as the ability of a chemical entity, e.g., a drug molecule, to permeate, i.e., to go through, a defined environment, e.g., biological or biomimetic barriers”. I suggest to correct and rewrite “In this context, it is also important to underline the role of efflux transporters that work against drug accumulation in the blood (Fig. 4(e))”.  There is no reference for the sentences “Amidon et al. correlated the rate of drug absorption across the intestinal epithelium with a parameter named human permeability, Ph. Ph was introduced as a derived coefficient calculated from of the human fraction absorbed, A%, of a drug compound”.  

Comment 6: In the sentence “Where jh is the human intestinal flux of the drug, which is obtained by normalizing A% by the estimated surface area of the gastrointestinal wall and ch is the drug concentration at the membrane, i.e., at surface of the intestinal wall”,? authors discussing about which drug?. The statement of 266-278 (page 9) must be justified by citation.     

Comment 7: In figure 5A, each part of Franz diffusion cell must be labeled. Moreover, I appreciate to write drawbacks of Franz diffusion cell.

Comment 8: In figure 15, the dissolution curves up to 50 min showed sigmoidal pattern. After 50 min, there is steady state. How did authors draw a linear patter of permeation after 50 min as shown in grey color?. What do you mean by new formulations?    

Comment 9: I suggest to provide detail source of “from Merck Millipore” with city, state, and country.

Comment 10: There are many publications published using EpiDerm, Start-M, and related in vitro models. I suggest to present major findings of reported model drug for topical and oral delivery. In table, model drug, permeation parameters, in vitro model and pros/cons must be summarized for recent findings (updated literature).  

Author Response

The authors are grateful for the valuable and insightful comments from the expert reviewer and for the time and effort, which helped to improve this comprehensive review. We have addressed all objective points raised by the reviewer. Alterations to the original manuscript text have been highlighted in green (or yellow) in case comments from different reviewers overlapped. We also took the chance to proof read the manuscript. Eventual alterations to the original text are highlighted in grey

Comment 1: Abstract must be short and concise. The used abbreviations must be defined appeared first time in the text body such as BCS, ICH, and ADME etc. There must be consistency in using “in vitro” (must be italic or non-italic). Copyright trade mark must be removed (PermeaPad®). In the sentence “……predicting the in vivo situation indicating that…”, define the in vivo situation? Please replace the sentence “generic drugs development” with “generic product development”. 

As requested by the reviewer, we have shortened the abstract. We have defined the abbreviations upon first usage.

We decided to use non italics for both “in vitro” and “in vivo” as this seems to be nowadays the trend. We have removed the trademark from the PermeaPad (except on first usage). To streamline the manuscript, we have removed all trademarks.

To clarify, we have replaced “in vivo situation” with “in vivo formulation performance” in the sentence “……predicting the in vivo situation indicating that…” (see page 1 line 19). Since we shortened the abstract considerably, the phrase “generic drugs development” was removed in the process.  

Comment 2: In the sentence “From 2016 to 2017, this number decreased slightly (-4.4%) [11]. Sadly, the slight decrease was followed by a significant increase by over 10% in 2018”, I found contradictory observation after comparing figure 2. Please check

We have checked Fig. 2 and we decided to remove the %-values from the text because the %-values vary according to which year we refer to. Therefore, to avoid confusion %-values were removed from the text but can still be found in the figure.

Comment 3: There must be a separate search methodology section along with inclusion and exclusion criteria.

Following the reviewer’s comment, we have created a separate sub-section “2.1 Methodology of the search” describing the search methodology (see page 8)

Comment 4: Page 7, line 211-225, I suggest to include reference source for the content. Table 1 must be corrected for the last column (percent correction required)

We have highlighted that the source is Scopus and we reported it in Table 1. We have also checked the correctness of the %-values.

Comment 5: Correct the sentence “the field of pharmaceutics”. Please provide a suitable reference against the sentence “Permeability can be defined as the ability of a chemical entity, e.g., a drug molecule, to permeate, i.e., to go through, a defined environment, e.g., biological or biomimetic barriers”. I suggest to correct and rewrite “In this context, it is also important to underline the role of efflux transporters that work against drug accumulation in the blood (Fig. 4(e))”.  There is no reference for the sentences “Amidon et al. correlated the rate of drug absorption across the intestinal epithelium with a parameter named human permeability, Ph. Ph was introduced as a derived coefficient calculated from of the human fraction absorbed, A%, of a drug compound”.

We have corrected the sentence containing “the field of pharmaceutics” into “was introduced in pharmaceutics” (see Page 12 line 229).

We have added the requested references.

As suggested by the reviewer, the sentence “In this context, it is also important to underline the role of efflux transporters that work against drug accumulation in the blood (Fig. 4(e))“ was rewritten into “In the context of active carrier-mediated transport, it is also important to underline the role of efflux transporters that decrease drug absorption by transporting the drug from the cell back into the lumen against the concentration gradient (see Fig. 4(a) E)” (see page 12)

Comment 6: In the sentence “Where jh is the human intestinal flux of the drug, which is obtained by normalizing A% by the estimated surface area of the gastrointestinal wall and ch is the drug concentration at the membrane, i.e., at surface of the intestinal wall”,? authors discussing about which drug?. The statement of 266-278 (page 9) must be justified by citation.     

We are referring to any drug which is absorbed through the intestinal wall. We have added the supporting references to the statements as requested.

Comment 7: In figure 5A, each part of Franz diffusion cell must be labeled. Moreover, I appreciate to write drawbacks of Franz diffusion cell.

The Franz cell parts in Fig. 5(a) are labelled with donor and acceptor, and the caption to Figure 5 is quite explanatory. As the review focuses on the barrier more than on the assay and set-ups, and the review is already quite long, we regard it as not applicable to insert additional information on the applicability of one specific diffusion cell over the other. However, since stirring influences the thickness of the unstirred water layer, we have added a comment on the stirring possibilities in vertical vs horizontal set-ups, which accommodates the reviewer’s comment (see page 13).

Comment 8: In figure 15, the dissolution curves up to 50 min showed sigmoidal pattern. After 50 min, there is steady state. How did authors draw a linear patter of permeation after 50 min as shown in grey color?. What do you mean by new formulations?

We thank the reviewer for this valuable comment. The small box is actually a dissolution profile and not a permeation profile as the figure could led to believe. We removed the dotted lines. As this is a poorly soluble but also poorly permeable compound, the drug is fully dissolved after approx. 1 hour but the permeation is much delayed and become significant only after this time. We changed the graphics in Fig. 15 for increased clearness.

Comment 9: I suggest to provide detail source of “from Merck Millipore” with city, state, and country.

We added the city and country related to Strat-M (Section 5.2)

Comment 10: There are many publications published using EpiDerm, Start-M, and related in vitro models. I suggest to present major findings of reported model drug for topical and oral delivery. In table, model drug, permeation parameters, in vitro model and pros/cons must be summarized for recent findings (updated literature).

We agree with the reviewer that the literature for Strat-M is larger than what we have reported. However, this review focuses on oral absorption. We decided to include also artificial barriers used for skin permeation studies to show the full portfolio of possibilities. As the review is already quite long, we feel it is better not to include additional information at this stage. We have not included EpiDerm in our review since it is a cell-based model (https://www.mattek.com/products/epiderm/), which is outside the scope of this review.

Round 2

Reviewer 2 Report

Authors revised the manuscript as per suggestion. Now, it is suitable for publication